# An adaptive threshold neuron for recurrent spiking neural networks with nanodevice hardware implementation

Ahmed Shaban [1], Sai Sukruth Bezugam [1] & Manan Suri [1✉]

We propose a Double EXponential Adaptive Threshold (DEXAT) neuron model that improves the performance of neuromorphic Recurrent Spiking Neural Networks (RSNNs) by providing faster convergence, higher accuracy and a flexible long short-term memory. We present a hardware efficient methodology to realize the DEXAT neurons using tightly coupled circuit-device interactions and experimentally demonstrate the DEXAT neuron block using oxide based non-filamentary resistive switching devices. Using experimentally extracted parameters we simulate a full RSNN that achieves a classification accuracy of 96.1% on SMNIST dataset and 91% on Google Speech Commands (GSC) dataset. We also demonstrate full end-to-end real-time inference for speech recognition using real fabricated resistive memory circuit based DEXAT neurons. Finally, we investigate the impact of nanodevice variability and endurance illustrating the robustness of DEXAT based RSNNs.

---

[1] Electrical Engineering, Indian Institute of Technology, Delhi, India. ✉email: manansuri@ee.iitd.ac.in

Neuromorphic Spiking Neural Networks (SNNs) are promising computational paradigms that take deep inspiration from the working of mammalian brains, firing neurons, and synaptic plasticity. SNN algorithms and specialized SNN hardware have been shown to be useful for addressing a wide variety of real-world data-centric applications. While conventional artificial neural networks (ANNs) primarily depend on continuous valued functions and supervised gradient descent based learning rules, SNNs also exploit sparse neuron spikes and unsupervised learning rules. ANNs have been shown to surpass SNNs and recurrent-SNNs (RSNNs) in terms of accuracy[1,2]. However, SNNs are relevant as they hold the promise for energy-efficient hardware realization owing to their bio-inspired nature[3]. Activation functions or neuron models play a pivotal role in the overall learning accuracy and energy efficiency of both ANN and SNN implementations. If RSNNs with spike- based temporal computation are to perform better on sequential tasks, it is essential that they get the capability of Long Short-Term Memory (LSTM) cells as stable working memory/memory states. In this regard, in a recent theoretical work[4] authors have shown that inclusion of Adaptive-Leaky Integrate and Fire (ALIF) neurons in RSNN can improve their computational capabilities. Such neurons are also used in implementing RSNN that can learn through hardware friendly algorithms like e-prop[5]. Adaptive neuron models generally incorporate biological property of neuronal adaptation[6] by either removing a part of membrane current after each spike event or by increasing the leak current[7–9]. However, in[4] authors demonstrate the use of an adaptive threshold model in which the firing threshold varies dynamically on spike event. The threshold voltage in such model increases on neuron firing and decays back exponentially to a baseline.

Hardware implementation of neuron models such as Integrate and Fire (IF) and Leaky Integrate and Fire (LIF)[10,11] have been widely reported in literature both with complementary metal oxide semiconductor (CMOS) and emerging resistive memory devices like resistive random access memory (RRAM), phase change memory, conductive bridge random access memory etc.[12–20] However, there are limited studies that show hardware implementation of adaptive/dynamically adaptive neuron functions. Mixed signal silicon implementation of adaptive neuron models exploiting membrane dynamics like Mihalas-Niebur model[9] and adaptive exponential decay model (AdeX)[8] have been shown in[21,22]. Wang et al.[23] have recently simulated a RRAM device to realize an adaptive neuron, in which the adaptation behavior is achieved by increasing the neuron membrane resistance at each spike event.

In this work our contribution is twofold; first, we propose a new dynamically varying Double EXponential Adaptive Threshold (DEXAT) neuron model for RSNNs, and second we show an efficient hardware implementation of the new neuron model. In the proposed DEXAT neuron model the threshold voltage decays by two exponential rates (slow and fast). We observe that the proposed model has several benefits while realizing RSNNs such as faster convergence, higher accuracy, re-configurability, and ease of hardware implementation. For hardware realization, we demonstrate the capability of non-filamentary OxRAM (oxide-based resistive memory) devices to realize double exponentials, using extracted data from multiple OxRAM material stacks. We also show experimental demonstration of the DEXAT neuron that exploits unique and tightly coupled circuit–device properties of non-filamentary OxRAM devices. Further, we perform system-level RSNN simulations on classification of sequential MNIST (SMNIST) handwritten digits and speech commands from GSC dataset based on the neuron parameters extracted from experiments. We demonstrate full end to end RSNN using fabricated resistive memory based DEXAT neurons for live speech recognition application on GSC dataset.

## Results

**Proposed adaptive neuron model.** Authors in[4] propose an architecture of RSNN called Long Short-Term Spiking Neural Network (LSNN) to enhance their computational performance. This involves providing RSNN with a LSTM by including ALIF neurons in the network. Hence, LSNNs consist of a network of LIF and ALIF neurons and can achieve similar performances as LSTM on sequential tasks. Threshold voltage adaptation in the ALIF neuron model is described by Eqs. (1) and (2).

$$B_j(t) = b_{j0} + \beta b_j(t) \tag{1}$$

$$b_j(t + \delta t) = \rho_j b_j(t) + (1 - \rho_j)z_j(t) \tag{2}$$

$B_j(t)$ is the adaptive firing threshold of the neuron, $b_{j0}$ is the baseline threshold voltage, $\beta$ is a constant scaling factor which scales the deviation $b_j(t)$ from the baseline $b_{j0}$, $\rho_j = exp(-\delta t/\tau_a)$ and governs the exponential decay of threshold voltage, $\tau_a$ is the adaptation time constant of the threshold decay and $z_j(t)$ denotes the output spike generated by neuron "j" having value 1/δt for a spike event and zero otherwise. The equations are described in discrete time with a minimum time step of δt. In this model the threshold voltage is increased by a fixed amount $\beta/\tau_a$ when the neuron fires and then decays exponentially back to a baseline value $b_{j0}$ with a time constant $\tau_a$ (see Fig. 1a).

We propose a new adaptive neuron model in which the threshold voltage decays with a double exponential having two time constants. Double exponential decay is governed by a fast initial decay and then a slower decay over a longer period of time. Threshold adaptation in our DEXAT neuron model is described by Eqs. (3), (4), (5).

$$B_j(t) = b_{j0} + \beta_1 b_{j1}(t) + \beta_2 b_{j2}(t) \tag{3}$$

$$b_{j1}(t + \delta t) = \rho_{j1} b_{j1}(t) + (1 - \rho_{j1})z_j(t) \tag{4}$$

$$b_{j2}(t + \delta t) = \rho_{j2} b_{j2}(t) + (1 - \rho_{j2})z_j(t) \tag{5}$$

where $\rho_{j1} = exp(-\delta t/\tau_{a1})$, $\rho_{j2} = exp(-\delta t/\tau_{a2})$. Equation (3) denotes the magnitude of threshold voltage at each time step of δt. First terms in Eqs. (4) and (5) for $b_{j1}(t)$, $b_{j2}(t)$ define the magnitude of threshold voltage decay governed by the two exponentially decaying factors $\rho_{j1}$, $\rho_{j2}$, and second terms govern threshold voltage increment when an output spike $z_j(t)$ occurs. The two equations get independently updated at each time step of δt. Values from both the equations are individually scaled by factors $\beta_1$, $\beta_2$ and added in (3) to define the magnitude of threshold voltage at time "t". $\tau_{a1}$ and $\tau_{a2}$ are two values of adaptation time constants governing the decay of threshold voltage. Here, $\tau_{a1}$ and $\tau_{a2}$ represent the small and large time constant respectively. Figure 1b shows qualitatively the changes in threshold voltage of neuron according to this model in response to firing.

Authors in[24] have demonstrated the benefits of an ALIF based LSNN using a STORE-RECALL task. STORE and RECALL is a delayed response task that tests the capability of a network for possessing "short-term" memory or a working memory. Working memory is defined as the ability to store and manipulate information over a short duration of time and forms the basis of temporal and cognitive processing[25]. We use the same task for comparing the performance of our DEXAT based LSNN to ALIF based LSNN. The task involves presenting the network with an input sequence consisting of two characters ("0" and "1") encoded by a spike sequence. The network has to store a character in its working memory on receiving a STORE instruction and has to recall the character on a RECALL instruction. The time duration between STORE and RECALL instruction is the working memory of the network. A LSNN network with 10 LIF and 10 ALIF

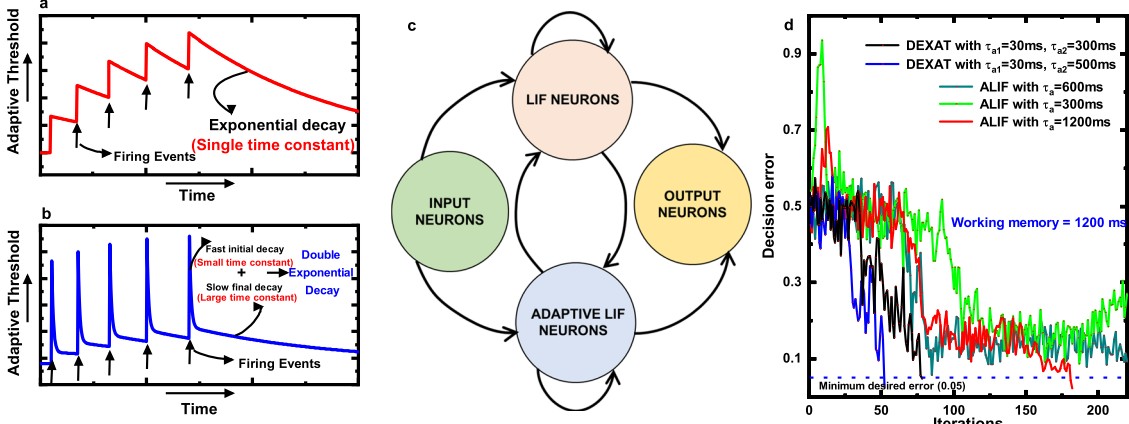

**Fig. 1 Qualitative description of neuron models and network architecture for LSNN tasks. a** Model in which neuron's threshold decays exponentially with a single time constant and (**b**) Proposed DEXAT model used in this work where threshold decays with two time constants. **c** LSNN Network used for STORE-RECALL task and (**d**) Learning curves of LSNN with ALIF and DEXAT adaptive neurons on a STORE-RECALL task for a working memory requirement of 1200 ms.

neurons as shown in Fig. 1c is used for this task. Time step $\delta t$ is taken to be 1 ms in simulations. The network is trained for 200 iterations with a minimum desired decision error of 0.05. Figure 1d shows the performance of LSNN with DEXAT and ALIF neurons for STORE and RECALL task requiring a working memory of 1200 ms. With ALIF neurons, the LSNN converges for $\tau_a$ of 1200 ms in 200 iterations while for smaller $\tau_a$ values of 600 and 300 ms LSNN network fails to converge to the desired decision error. On the other hand, LSNN with DEXAT neurons reaches the decision error requirement successfully even with adaptation time constants $\tau_{a1} = 30$ ms, $\tau_{a2} = 300$ ms (i.e., much smaller than the working memory requirement of 1200 ms). A case for a working memory of 600 ms is also presented for different time-constant values of ALIF and DEXAT neurons (Supplementary Fig. 1). We perform multiple simulations with different ranges of working memory requirements and adaptation time constants as shown in Fig. 2 to investigate their relation for optimum performance. In Fig. 2a we vary the adaptation time constant ($\tau_a$) of ALIF neuron to observe the ranges of working memory it can support. We conclude from Fig. 2a that the performance of LSNN with ALIF neurons is optimum when the adaptation time constant $\tau_a$ is greater than or equal to the value of required working memory. In Fig. 2b we fix smaller time constant $\tau_{a1}$ to 30 ms and vary larger time constant $\tau_{a2}$ of DEXAT neuron. We observe that for a case of $\tau_{a2} = 200$ ms and working memory requirement 2000 ms (i.e., working memory/$\tau_{a2} = 10$) also, LSNN converges successfully within 200 iterations using DEXAT neurons; not possible with ALIF neurons as shown in Fig. 2a. Hence, DEXAT neurons provide a LSNN the capability to converge to a low required decision error even when adaptation time-constant value is much smaller than the required working memory. In Fig. 2c we fix larger time constant ($\tau_{a2}$) and vary smaller time constant ($\tau_{a1}$) for different working memories. We observe from Fig. 2c that optimum performance is obtained even for large working memories if the ratio $\tau_{a2}/\tau_{a1}$ is greater than six. However, with a $\tau_{a2}/\tau_{a1}$ ratio of three the network fails to converge for larger working memories (i.e., for 2000 and 2400 ms). We also test the effect of scaling parameters (see Eqs. (3), (4), (5) in our model by varying the ratio $\beta_2/\beta_1$ as shown in Fig. 2d. It is observed that performance remains unaffected for a wide range of working memory size. Hence, from Fig. 2b, c, d we obtain a design space for tuning parameters used in our neuron model for best performance as shown in Fig. 2e. Thus, LSNNs with DEXAT neurons can support working memories which are much larger

than the corresponding neuron adaptation time constants compared to LSNNs with only ALIF neurons. From hardware implementation point of view, this relaxes the requirements from on-chip devices/circuits that are used for providing re-configurability to generate a range of time constants for sequential tasks with different working memory requirements. Further, in all the cases LSNN network with DEXAT neurons takes lesser number of iterations to converge compared to that consisting of ALIF neurons which can also prove beneficial.

We discuss the reasons for better performance of proposed DEXAT model in Supplementary Note 1 and Supplementary Note 2. Presence of two time constants (i.e., a fast decaying exponential followed by a slow decaying exponential) in the adaptive threshold of neuron enables the network to fine tune weights during training thus achieving higher accuracy. Further, proposed DEXAT model provides a neuron spike-activity dependent (spike frequency) weight update mechanism. This enhances learning of individual neurons and results in faster network convergence.

**Hardware implementation.** Emerging nonvolatile memory (NVM) devices have been used to realize multiple synaptic and neuronal functions. Most neuromorphic NVM implementations in literature primarily rely on conductance modulation of the devices. For instance, analog multilevel cell (MLC) characteristics are exploited for realizing synaptic weight update characteristics like Long Term Potentiation (LTP) and Long Term Depression (LTD) by increasing or decreasing the device conductance respectively. For efficient hardware realization of the proposed DEXAT neuron, multiple nonlinear/analog functions need to be harnessed. In particular: (i) we need a circuit–device property that can help realize the double exponential with varying time constants, during the decaying phase of the threshold and (ii) another property that helps to increase (adapt) the peak threshold value after each consecutive firing event. We propose to extract both of these analog properties from the circuit–device interactions of OxRAM devices in the following manner:

- Asymmetric conductance change: Non-filamentary OxRAM devices in which resistance change is determined by modulation of the interfacial Schottky/tunneling barrier by the electron trapping/detrapping or ion migration have potential for MLC characteristics[26]. Pulse characterization of these devices used as synapses show asymmetric relative

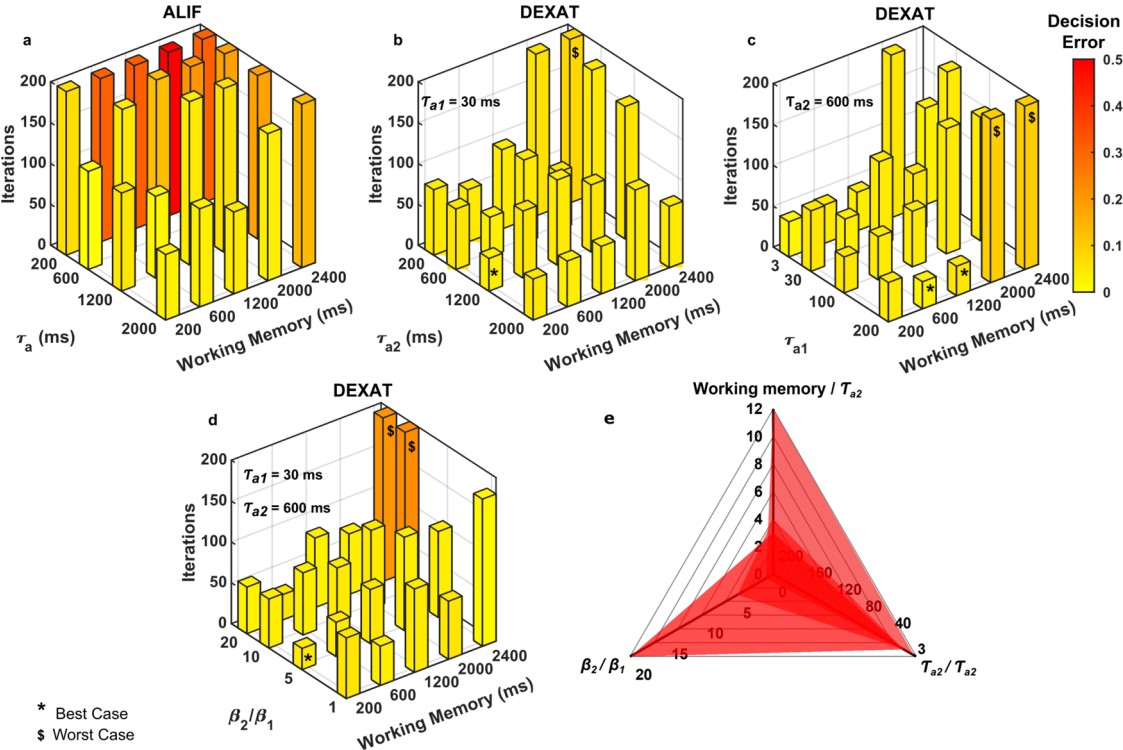

**Fig. 2 Performance analysis of ALIF and DEXAT neurons. a** Performance of LSNN on STORE-RECALL task with ALIF neurons and $\tau_a$ varied for different working memory requirements. Performance of LSNN on STORE-RECALL task with DEXAT neurons and (**b**) $\tau_{a1}$ fixed at 30 ms and $\tau_{a2}$ varied for different working memory requirements (**c**) $\tau_{a2}$ fixed at 600 ms and $\tau_{a1}$ varied for different working memory requirements (**d**) $\tau_{a1} = 30$ ms, $\tau_{a2} = 600$ ms and ratio $\beta_2/\beta_1$ varied for different working memory requirements (Shorter and yellower bars denote faster convergence and lower decision error, taller, and redder bars denote non-convergence and higher decision error). **e** Radar plot showing design space for tuning parameters in DEXAT model for obtaining best performance (For optimum performance, the region formed by the three set of points should lie within the shaded region).

conductance change between two or more consecutive LTP/LTD pulses[27–33]. Such relative conductance change nonlinearity may be attributed to the inherent diffusion/drift dynamics of the ions/vacancies involved in switching[32]. In such devices after reaching the peak LTP state, abrupt or very high conductance drop on application of the first (or initial few) RESET pulse(s) has been commonly observed. The relative change in conductance on application of subsequent RESET pulses (i.e., after the first few) is more gradual and tends to saturate. This nonlinear behavior is considered undesirable while emulating ideal linear synaptic characteristics, and efforts have been made to overcome the nonlinearity through use of complex pulse schemes[34]. However, we exploit this undesirable combination of initial abrupt conductance jump followed by subsequent gradual conductance jumps, to realize the dual time constants (i.e., faster and slower decaying exponential respectively) in our proposed DEXAT neuron model.

- Coupling of OxRAM initial resistance state with programmed resistance state: We exploit the coupled nature of initial and final programmed states of OxRAM to realize the adaptive behavior of the threshold with consecutive firing events. The programmed conductance state of an OxRAM device would depend on two factors: (i) the programming condition used (i.e., pulse–voltage, duration) and (ii) the initial conductance state prior to application of programming pulse. In proposed DEXAT hardware realization the programming SET pulse on each firing event is identical, however the initial high resistance state (HRS) prior to the application of the SET pulse is variable. Our programming methodology (detailed in next section)

ensures that the initial HRS is a function of the time between two consecutive firing events. If the time between consecutive firing events is less the RESET pulse is shorter and thus the corresponding initial HRS is weak. Similarly, if the time between two consecutive firing events is larger the devices ends up in a stronger HRS state owing to the application of a longer RESET pulse. Based on the desired biological adaptation of firing threshold, if two consecutive firing events occur within a short time duration the subsequent firing threshold of the neuron must increase and should make the second firing event harder to achieve. This same effect is emulated in our implementation as when HRS is weaker the SET pulse leads the OxRAM device in a higher conductance state compared to the case when the HRS state was stronger. Using this phenomena and the proposed programming methodology we are able to ensure progressively increasing firing threshold after every spike event without modifying the intermediate SET condition (see below).

For realizing adaptive threshold voltage, we translate the device conductance changes arising due to the above two properties using an appropriate circuit described in the following section. We validate our proposed DEXAT methodology based on experimental data extracted from multiple different non-filamentary OxRAM device stacks. We use the experimentally observed typical LTP-LTD curves from four different devices showing asymmetric conductance change[27–29] (Fig. 3a–d). LTD part of the curve provides the two types of asymmetric jumps for emulating the dual decaying adaptive thresholds (and estimating the respective decay time constants). Discrete LTD pulses on X-axis are upsampled using bi-linear interpolation choosing a

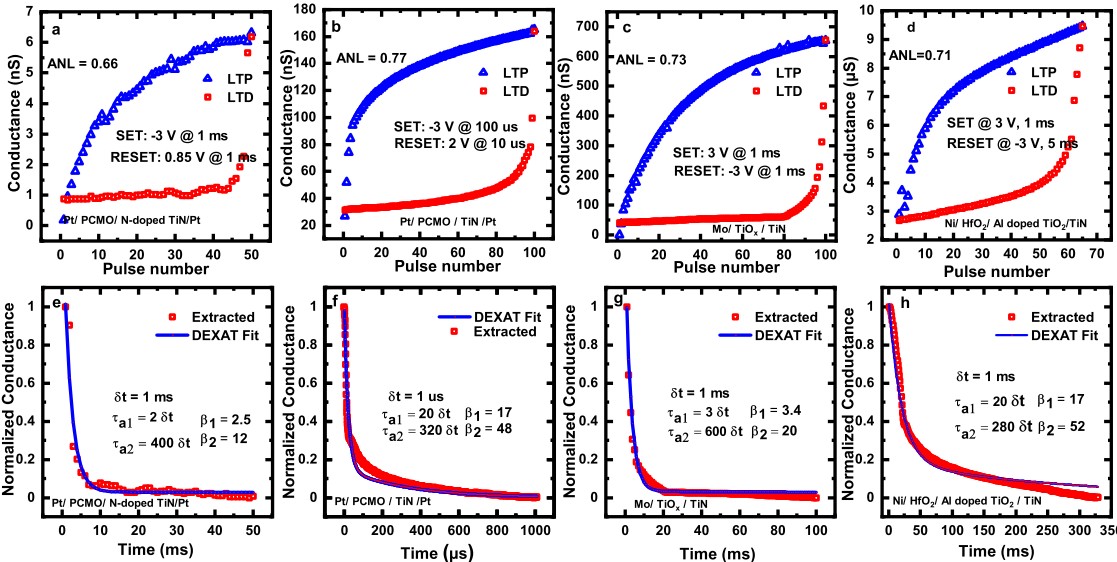

**Fig. 3 DEXAT behavior extraction from non-filamentary OxRAM devices. a–d** Extracted LTP-LTD characteristics of Pt/PCMO/N-doped TiN/Pt, Pt/PCMO/TiN/Pt, Mo/TiOx/TiN, and Ni/HfO₂/Al-doped TiO₂/TiN devices respectively. **e–h** Normalized and interpolated LTD conductance curves corresponding to (**a–d**), respectively, fitted with DEXAT neuron equations. Extracted DEXAT neuron parameter values are indicated inside respective curves.

suitable sampling time "$\delta t$", which is equivalent of the LTD curve showing the effect of applying a long duration RESET pulse. Conductance values are normalized between 0 and 1 using min–max normalization. The resultant conductance curves are then fitted with DEXAT equations to extract values of time constants $\tau_{a1}$ and $\tau_{a2}$ and scaling factors $\beta_1$, $\beta_2$ shown in Fig. 3e–h. We calculate the measure of asymmetric nonlinearity (ANL) as in[33] for the four device stacks. ANL value lies between "0" and "1", where "0" denotes a fully symmetric case. All four types of devices exhibit a high value of ANL (see inset of Fig. 3a–d). In order to achieve double exponential time constants a high ANL value is helpful, contrary to existing efforts reported in literature, that try to reduce the value of ANL for improving traditional synaptic emulation.

Out of the four stacks considered for this study, we perform full DEXAT circuit experiments on the bilayer device with Ni/HfO₂/Al-doped TiO₂/TiN stack (Fig. 3d, h) as described in the following section. DC characterization of this device is shown in Supplementary Fig. 4 with fabrication details presented in "Methods".

Figure 4 shows the functional block diagram, circuit, and programming methodology for the hardware realization of our proposed DEXAT neuron. Peripheral circuit blocks such as integrator, comparator, pulse generator have been extensively investigated in existing literature[19,21–23,35] and can be realized using standard circuit elements (CMOS, capacitors etc.). Our main focus in this section is efficient realization of the proposed novel threshold adaptation block with capability to realize double exponential decay behavior inside the DEXAT neuron. The neuron fires when output of integrator i.e membrane potential ($V_{mem}$) crosses the voltage on threshold terminal ($V_{th}$) of comparator. Firing triggers the pulse-generator block to generate a sequence of control pulse signals (and their complements) which activate the SET and RESET paths in the threshold modulator circuit as shown in Fig. 4b–d. Figure 4a also shows the control signals generated on a firing event which are fed to the threshold modulator block in order to control the different modes of the circuit. In our proposed programming scheme, on the occurrence of each spike event, a fixed short-duration SET pulse followed by a variable duration RESET pulse is applied to the OxRAM device

(Fig. 5). The exact duration of the RESET pulse depends on the time after which the next consecutive spike occurs. The neuron circuit requires an initialization phase before starting to integrate the incoming input spikes. When an output spike occurs the OxRAM device first undergoes a SET process. During this period transistors MN1 and MP4 turn ON and form the SET path as shown in Fig. 4b while transistors MP1, MP3 and MN2 are turned OFF. Although transistor MP2 is ON during this period as $V_R$ is low, its effect on SET process is negated by the OFF MP3 transistor. The RESET path of the device is then activated through the ON MP1, MP3 and MN2 transistors as shown in Fig. 4c while all other three transistors are OFF. The threshold voltage decays during the duration of application of the RESET pulse. Greater the time difference between the next spike more will be the decay of the threshold voltage. The proposed circuit is event driven and all transistors except MP2, MP3, and MN2 are turned OFF when the threshold voltage saturates (i.e., once the baseline voltage is reached) in the absence of a spike event. A reduced supply voltage is applied on the device in IDLE mode to reduce the stress on the device. Further, a smaller voltage also ensures that the resistance state of the device is not disturbed while in IDLE state. The voltage at Node "D" is governed by the ratio of final OxRAM resistance state and the passive resistor (shown in Fig. 4d).

Figure 5 shows experimentally observed traces for an example spike-sequence that we label as S1 for reference. Figure 5a denotes output spikes or occurrence of neuron firing. Figure 5b shows the programming signals applied to the OxRAM when the pulse-generator module feeds control pulses to the threshold modulator circuit. The pulse sequence consists of a fixed duration SET pulse and a variable duration RESET pulse, where duration of the RESET pulse depends on the time interval between occurrence of consecutive output spikes. As soon as an output spike is generated it initializes the pulse-generator block shown in Fig. 4a; the previous RESET pulse is terminated and a new sequence of post spike SET/RESET pulses is generated. In Fig. 5b RESET pulse duration is 50 ms for first three spikes events. The threshold voltage peak that appears immediately after the end of each SET pulse or the onset of RESET pulse (shown in Fig. 5c) is a consequence of the low-resistance state attained by the device due to the SET pulse. After every local threshold peak value, first a fast

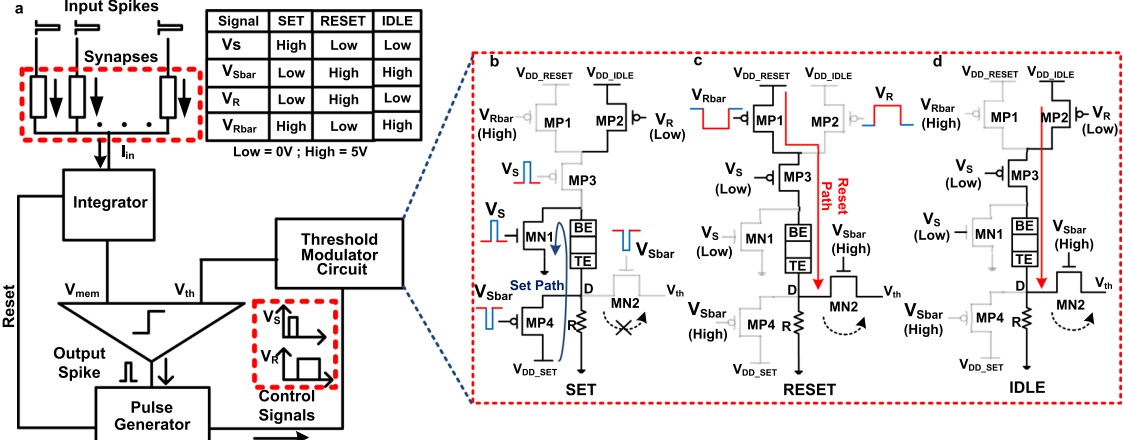

**Fig. 4 Proposed DEXAT neuron threshold modulator. a** Functional block diagram of proposed adaptive neuron and associated control signals generated by pulse generator in a spike event and (**b**), (**c**), (**d**) SET, RESET, and IDLE modes in our proposed 6T-1R threshold modulator circuit respectively.

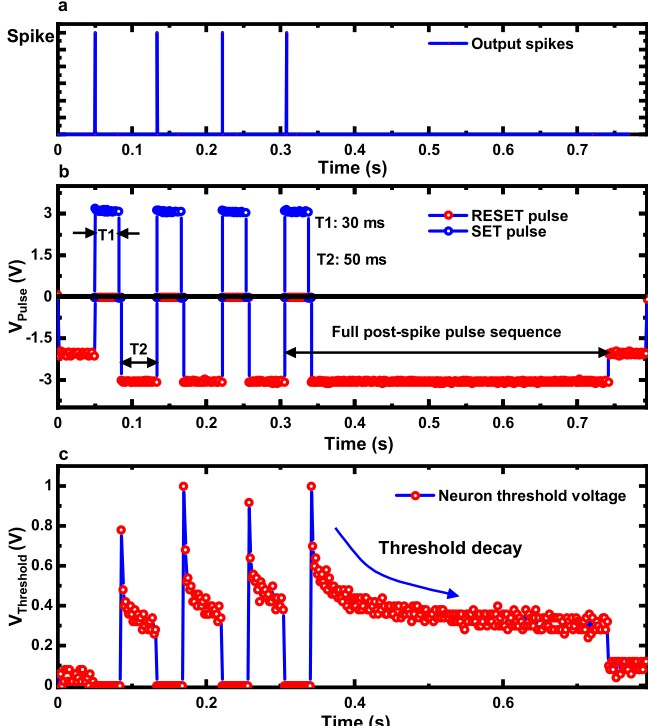

**Fig. 5 DEXAT threshold modulator experiment. a** Output spikes of the neuron denoting a sequence S1 of four firing events. **b** Input pulses applied to OxRAM device in threshold modulator circuit initiating threshold modulation in case of a spike event. A 3 V, 30 ms SET pulse ($V_{DD\_SET} = 3$ V) applied at top electrode (TE) is followed by a 3 V, 50 ms RESET pulse ($V_{DD\_RESET} = 3$ V) applied at bottom electrode (BE) after first spiking event. A voltage of 2 V ($V_{DD\_IDLE}$) is applied on BE after threshold voltage saturates in absence of firing. (Voltages applied on BE are shown negative only for representation). **c** Experimentally observed threshold voltage of hardware neuron showing increase in threshold at each spike event and subsequent decay afterwards.

decay, followed by a gradual decay of the threshold is observed, during the course of the applied RESET pulse. This fast and gradual decay occur due to the asymmetric conductance change behavior of our device described in the previous section. The continuously applied/long RESET pulse on the OxRAM device has a cumulative effect similar to that of applying multiple short

RESET pulses. OxRAM's initial resistance as seen by consecutive SET pulse in the $(i + 1)_{th}$ spike-event depends on the final HRS reached due to the preceding RESET pulse in the $i_{th}$ event. A weak/shorter RESET pulse due to shorter inter-spike delay ensures that the final high resistance reached prior to $(i + 1)_{th}$ firing event is lesser compared to the state achieved prior to the $i_{th}$ spike event. Thus the SET pulse in $(i + 1)_{th}$ spike event drives the device to a stronger set state (as it started from a weaker initial HRS) compared to the SET pulse in $i_{th}$ event. Thus, the desired threshold adaptation behavior for a sequence of spike events is realized, as shown in Fig. 5c. If there is no spike event for a long duration, the threshold voltage continues to decay and finally saturates as shown in Fig. 5c. This emulates the adapting threshold model described by Eqs. (3), (4), (5).

Figure 6a shows the increment in neuron threshold at each spike event for sequence S1 and its fitting obtained using the DEXAT neuron model. The duration of SET pulse is omitted for the purpose of fitting. We observed cycle to cycle (C2C) variation in threshold voltage increment at each spike event as shown in Fig. 6b for multiple cycles. To further validate the functionality, an alternate spike sequence S2 was also experimentally tested on the OxRAM-based DEXAT circuit. Figure 6c shows the threshold voltage adaptation for S2 and corresponding fitting parameters obtained using DEXAT model. The effect of OxRAM device behavior is evident from Fig. 6d which shows that the magnitude of increment in threshold voltage (Δ) at each event starts to decrease with increasing number of spike events. The maximum value of threshold voltage attained in our neuron as a result of adaptation is experimentally observed to be ~1.2 V and is dependent on the OxRAM device's minimum resistance state. The minimum value of threshold (i.e., lowest decayed value obtained after application of RESET pulse) was experimentally found to be ~280 mV in our case. Our adaptive neuron provides an adaptation voltage range of about 5× the lowest value.

We also realize a fully digital implementation of the proposed threshold modulator circuit using FPGA and simulated digital ASIC hardware blocks as detailed in Supplementary Note 3. In Supplementary Note 4 we present a comprehensive benchmarking of our digital and memristive realizations of DEXAT with other state of the art adaptive neurons in literature.

## Learning results for LSNN with proposed DEXAT neuron
We use parameters extracted from proposed hardware neuron to simulate a standard benchmark test of classifying SMNIST

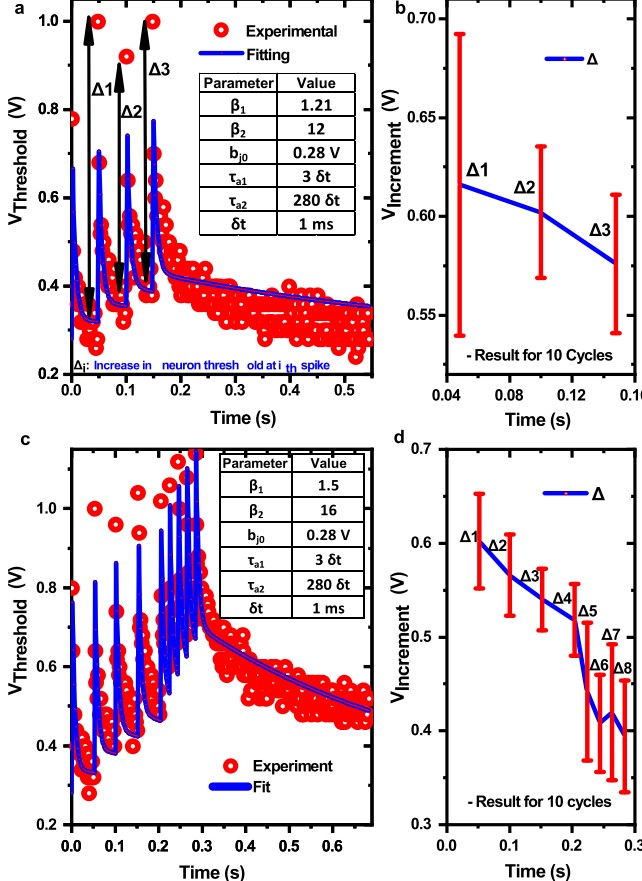

**Fig. 6 Hardware DEXAT neuron parameter extraction. a, c** Experimentally observed adaptive threshold behavior for spike sequence S1 and S2 respectively showing increment in threshold voltage (Δ) at each spiking event and fitting using DEXAT mode. Extracted parameters for both sequences S1 and S2 are shown in inset of graphs. **b, d** Variation in threshold voltage increment (Δ) for sequence S1 and S2 respectively at each spike event due to C2C device variability and decrease in threshold voltage increment (Δ) with subsequent spikes due to device behavior. Error bars represent standard deviation.

requiring LSTM. LSNN network used for this task is shown in Fig. 1c. Figure 7 shows the results when a digit is sequentially presented after training the LSNN network using our hardware neuron parameters. Figure 7b, c shows the spike rasters of LIF and DEXAT neurons respectively used in the network during the learning process. Figure 7d shows the dynamics of the firing thresholds of a sample of eleven DEXAT neurons in the network. It is evident that the adaptive nature of neurons provides a LSTM by virtue of large adaptation time constant and helps in achieving high accuracy for the sequential task at hand. An additional input neuron is used to generate output from the network. This neuron becomes active after the presentation of all 784 pixels of the input image. Firing probability of this neuron is shown in the top right corner of Fig. 7a. The softmax of 10 linear output neurons is trained using back propagation through time (BPTT)[36,37] to produce the label of the sequentially presented handwritten digit. This is shown in Fig. 7e by a yellow shading denoting the output label detected post testing. Figure 7f shows that during the intermediate duration of presenting the input image, probability of desired label varies but becomes maximum for correct inference by the end of 840 ms. In this case the input label corresponding to digit 4 is correctly detected. Figure 8a shows the test

accuracy benchmarking of the simulated LSNN using our hardware DEXAT neuron with networks comprising of other neuron/activation functions. The LSNN network using our hardware DEXAT neuron parameters achieves a test accuracy of 96.1% which is 2.8% higher than the ALIF based LSNN network reported in[4] and only 2.4% less than LSTMs.

We also benchmark through simulations the performance of LSNN networks based on DEXAT neurons realized using different OxRAM technology stacks (Fig. 3e–g). Baseline voltage in all three cases is taken as 0.28 V. Accuracy of 94.35%, 95.4%, and 96.1% for[27–29] was obtained, respectively, as shown in Fig. 8b. An important point to note is that despite the variation in the time constants for different devices the accuracy is almost unaffected for classifying SMNIST. We also perform network simulations after incorporating OxRAM device variability (see Methods for variability extraction methodology) in the DEXAT neurons. We analyze the network behavior over a wide range of $\eta_r$ values (0 to 40%) as shown in Fig. 8c and Supplementary Fig. 7. Even for the extreme case of $\eta_r = 40\%$, the accuracy drop was found to be only ~2.1% in case of SMINIST application shown in Fig. 8c proving the network and neuron's robustness towards both cycle to cycle (C2C) and device to device (D2D) variability. Table 1 shows the comparison of our proposed adaptive neuron with other reported works on adaptive neurons. We present estimated performance (energy, power, and area) of different implementations of our DEXAT neuron's threshold modulation block in Supplementary Note 4. It is important to note that OxRAM device will undergo programming (SET/RESET) during the inference process. This imposes high endurance requirements on the OxRAM devices being used to realize the neuron circuit. In Supplementary Note 5, we present a detailed analysis on the impact of OxRAM device endurance (resistance window degradation) on overall network inference accuracy.

In order to investigate advantages of the proposed DEXAT based LSNN for real-world temporal applications, we trained multiple LSNNs with varying dimensions, on the 12-class GSC dataset (see Supplementary Fig. 15 for LSNN benchmarking results). We found that LSNNs with the proposed DEXAT model match state-of-the-art accuracy (~91%) on the GSC 12-class dataset[38] while significantly outperforming literature in terms of total network resources (i.e., with 50 to 70% reduction in number of hidden layer neurons). DEXAT based LSNNs were also found to beat their ALIF based counterparts in lesser number of iterations (Supplementary Fig. 15b). Further, we designed an experimental setup (see Methods) as shown in Supplementary Fig. 9 and Supplementary Fig. 10 to demonstrate full end to end learning process inclusive of hardware DEXAT neurons for the GSC (2 class) real-time speech recognition application. Figure 9 shows the learning result for a real-time speech sample using the end to end experimental setup. The real-time spoken speech sample "UP" is preprocessed (see Methods for details) as shown in Fig. 9a. This preprocessed sample is then parsed to the network based on which spiking in the input neurons occurs as shown in Fig. 9b. Figure 9c shows the spike rasters for the hidden layer neurons during the speech recognition process. Figure 9d, e shows the adaptive thresholds of the DEXAT neurons defined in software and hardware respectively during the inference. Figure 9f shows the decision generated at the output layer with time. At the end of the inference the output decision value corresponding to "UP" is higher than that corresponding to "DOWN" signifying a correct decision. Movie of experimental demonstration showing live spoken speech recognition using real OxRAM-based hardware DEXAT neuron circuits can be accessed through Supplementary Movie 2. We also performed multiple cycles of experiments for input samples randomly chosen from GSC dataset corresponding to the two classes "UP" and "DOWN". The results are shown in Supplementary Fig. 11a–f.

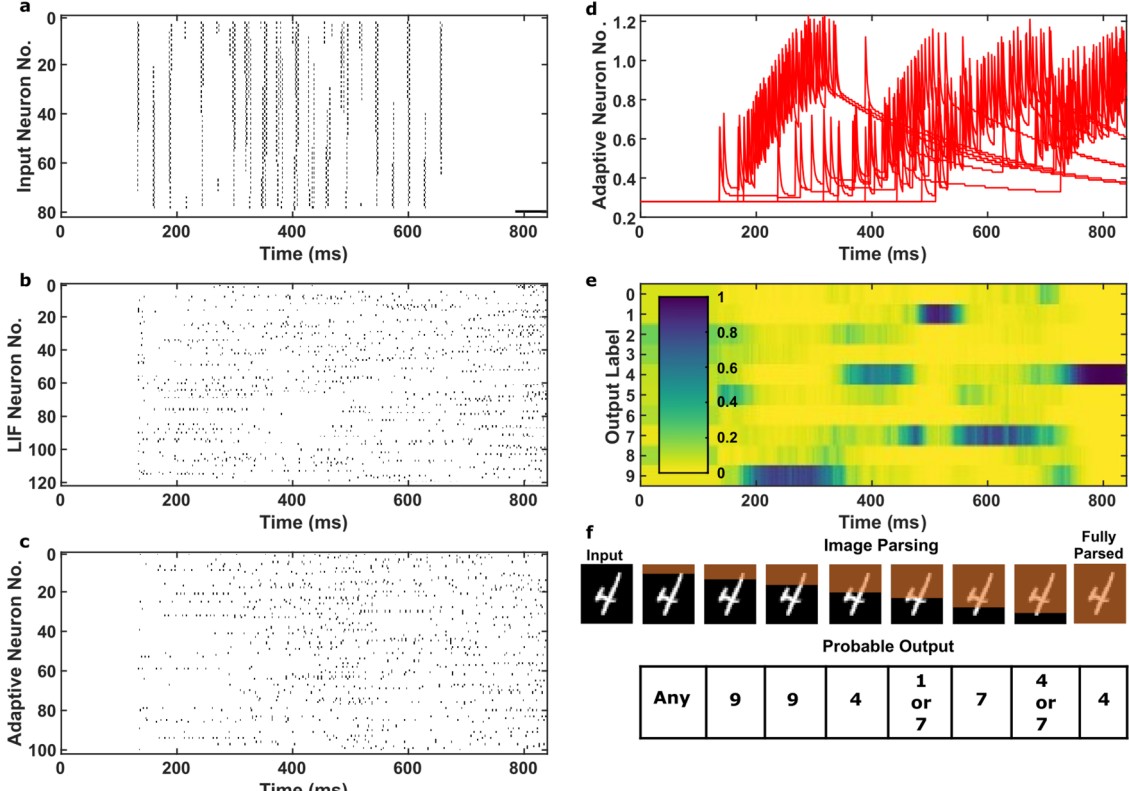

**Fig. 7 Dynamics of LSNN network for sequential MNIST obtained after training the network. a** Spike rasters from input neurons. **b**, **c** Spike rasters of LIF and DEXAT neurons respectively. **d** Dynamics of the firing thresholds of adaptive neurons. **e** Activation of softmax readout neurons. **f** Input test image presented sequentially pixel by pixel row-wise. The network gradually infers the image over a duration of 840 ms (Movie/animation of these results for better understanding can be accessed through Supplementary Movie 1).

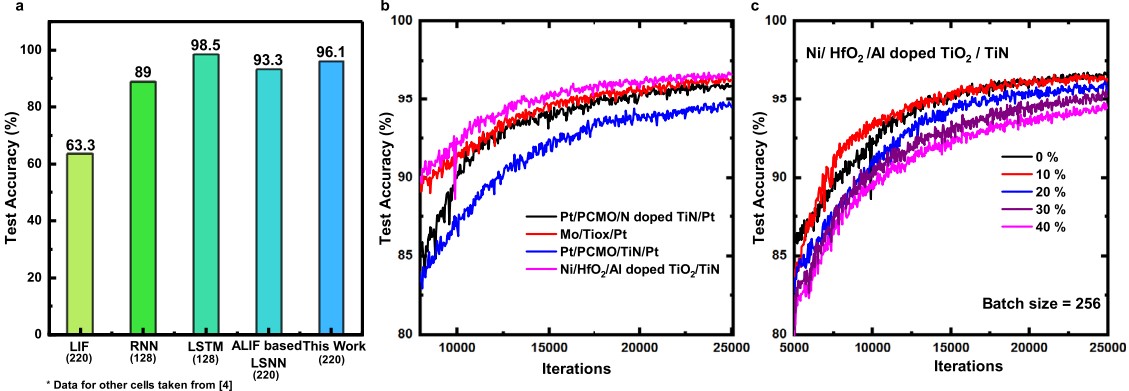

**Fig. 8 Benchmarking of DEXAT based LSNN. a** Test accuracy comparison for classifying SMNIST using different neuron cells. Numbers in brackets indicate number of cells. **b** Test accuracy comparison on SMNIST for parameters extracted from different device stacks and full DEXAT experiments on $HfO_2/TiO_2$ device. **c** Test accuracy comparison on SMNIST with different resultant variabilities for $HfO_2/TiO_2$ device.

## Discussion

Recent works have shown that the performance of neuromorphic RSNNs can improve by including adaptive LIF neurons. In this work, we propose a new DEXAT neuron model for use in RSNN. Our proposed model provides drastic benefits like higher accuracy, faster convergence and flexibility in hardware implementation compared to existing ALIF model. Benefits of proposed DEXAT based LSNN are shown on three diverse tasks (store-recall, SMNIST, and speech recognition). System-level simulations of LSNN network with DEXAT neurons achieved a test accuracy of 96.1% for classification on SMNIST dataset and 91% on GSC dataset. In case of GSC dataset, DEXAT based LSNNs were found to match state of the art accuracy

with significantly less number of hidden layer neurons. Further, we experimentally demonstrate that the proposed DEXAT neuron model can be realized in hardware using tightly coupled circuit–device interaction of different bilayer OxRAM devices (through hybrid CMOS-OxRAM circuits). Effects of OxRAM device variation and endurance on network performance are also investigated. LSNN networks with DEXAT neurons are found to be robust to OxRAM device variability and are able to achieve an accuracy of 94% with even 40% resultant device variability. We demonstrate real-time speech inference using full end-to-end experimental setup consisting of real hardware CMOS-OxRAM DEXAT neurons. Further, we explore digital implementations of the adaptive threshold

**Table 1 Comparison with other implementations of adaptive neuron models.**

| Ref. | Neuron Type | Threshold Adaptation Model/Principle | Implementation | Emerging Memory Used | Adaptation Circuit | Classification Task (Accuracy) | Learning Algorithm |
|---|---|---|---|---|---|---|---|
| [21] | Adaptive LIF | Mihalas Neibur Model | Experimental | None | 8T + 1 capacitor +3 switched capacitors | - | - |
| [22] | Adaptive LIF | Adaptive Exponential (ADeX) Model | Experimental | None | 3 MUX + 2 buffers +1 tunable resistor +1 OTA +1 capacitor | - | - |
| [23] | Adaptive LIF | Membrane Resistance Modulation | Simulation | RRAM | +7 switches 2 RRAM devices | Partial MNIST (10 images) (100%) | Unsupervised STDP |
| This Work | Adaptive LIF | DEXAT Model | Experimental | Bilayer OxRAM | 6T + 1 OxRAM device | 1. SMNIST (96.1%)[a] 2. Speech Recognition (~91%)[b] (GSC dataset for 12 classes) | BPTT |

(Detailed performance benchmarking is presented in Supplementary Notes 4, 5).
[a]Obtained a best accuracy run with 96.4%.
[b]Obtained a best accuracy run with 91.3%.

modulator block of the proposed DEXAT neuron for performance benchmarking. Proposed new DEXAT neuron model and its different hardware implementations have the potential to realize efficient large-scale recurrent/temporal neuromorphic networks. Among various realizations, hybrid CMOS-OxRAM circuits offer clear area-benefit. Future efforts should be in the direction of reducing individual OxRAM device switching energy to sub pico-joules i.e., through lower programming voltages, switching currents (~nA or lower), and ultra-fast switching speeds (sub ns)).

## Methods

**Experimental setup for basic characterization.** Characterization of the OxRAM devices is performed using Keithley 4200 SCS parameter analyzer. Keithley 4210 high power SMU (Source Measure Unit) was used to perform the DC sweep measurements. For DC characterization a dual voltage sweep is applied on TE using SMU with SET voltage from 0 to 3 V and a RESET voltage from 0 to $-5$ V. Pulse characterization of our device is performed by applying multiple SET and RESET pulses Keithley 4225 PMU and reading the device conductance after every pulse. We built our proposed neuron circuit on a General Purpose Board (Supplementary Fig. 5). CD4007UB CMOS dual complementary pair plus inverter IC is used for MOS transistors. The IC has enhancement type high voltage CMOS devices. An OxRAM device IC is used for the bilayer analog device. Firing events are emulated by giving a serial input to arduino microcontroller that acts as a pulse-generator block and outputs the desired control signal pulses denoting a firing event. Sequence of firing events can be varied by controlling the timing interval of serial inputs to microcontroller. Optimized SET and RESET voltages are provided by external power supply. The threshold voltage output is observed using a high resolution digital storage oscilloscope.

**Experimental setup for end to end speech recognition.** For the end to end real-time speech recognition experiments we fabricated a custom designed parent printed circuit board (PCB) having various blocks like microcontroller, analog to digital converter (ADC), digital to analog converter (DAC), and a separate daughter PCB consisting of DEXAT threshold modulator circuit components. Daughter PCB is interfaced with the parent PCB as shown in Supplementary Fig. 10. Both the PCBs together are used to emulate the complete LSNN network including multiple hardware DEXAT neurons. All LIF neurons are realized in software, while the DEXAT neurons are partitioned between hardware and software. Hardware DEXAT neurons communicate with their software counterparts as shown in Supplementary Fig. 9 (peripheral neuron circuit blocks like integrator, comparator are implemented in software, while adaptive threshold block is implemented using multiple CMOS-OxRAM circuits). Inference takes place temporally over time as each preprocessed input speech sample is presented in time step "$\delta t$" where $\delta t = 10$ ms. During the real-time inference, a serial input is synchronously fed to the PCB whenever a hardware DEXAT neuron in the network spikes. As a result, the control programming signals are applied to the hardware DEXAT neurons. The real-time hardware adaptive thresholds are transmitted to the ADC on board and is provided to the network code in real-time as shown in Supplementary Fig. 9. Tektronix MDO3024 Mixed Signal Oscilloscope is used for observing the real-time adaptive thresholds. For the hardware DEXAT neurons OxRAM device variabilities are inherently included in the network in real-time while extracted variability is incorporated in software neurons.

**Device fabrication.** Analog resistive switching OxRAM stacks of $Ni$/3 nm $HfO_2$/7 nm Al-doped-$TiO2(ATO)$/$TiN$ (top to bottom) structure were fabricated by following a CMOS compatible process in Prof. Tuo's Lab (NCTU). The active device area was 50 μm × 50 μm, and the ATO as well as $HfO_2$ were deposited using plasma-enhanced atomic layer deposition (PE-ALD). The device fabrication flow is as follows: first, 100 nm thick TiN BE flm was deposited on thermal-$SiO_2$ (500 nm)/ Si wafer by physical vapor deposition, RF magnetron sputtering. The BEs were then patterned by optical photolithography (first mask) and dry etching using inductively-coupled plasma (ICP). The bottom, 7 nm thick ATO dielectric, was then deposited by interchanging varying amount of $TiO_2$ and $Al_2O_3$ PE-ALD cycles, using TDMATi (Tetrakis(dimethylamido)titanium) and TMA (trimethyla-luminum) as metal-organic precursors and $O_2$ plasma as a reactant. Upper, 3 nm thick dielectric $HfO_2$ flm, was deposited using TDMAHf (Tetrakis(dimethylamido) hafnium) and $O_2$ plasma. All depositions were carried out at 250 °C using Veeco-CNT Fiji F202 remote plasma hot-wall reactor PE-ALD system. Top Electrode (TE) pattern (similar to the BE pattern but rotated 90°) was defined using second optical photolithography mask and 100 nm thick Ni TE film was deposited by DC sputtering and patterned using lift-off technique. Final photolithography (third mask) and ICP dry etching step was performed to open the contact windows (etch the dielectrics) to the BE contact pads. Wire bonding and packaging were the final steps for the OxRAM encapsulation.

**Resultant device variability extraction methodology.** In all the full network analysis we incorporate variability on the adaptive threshold voltage of each neuron

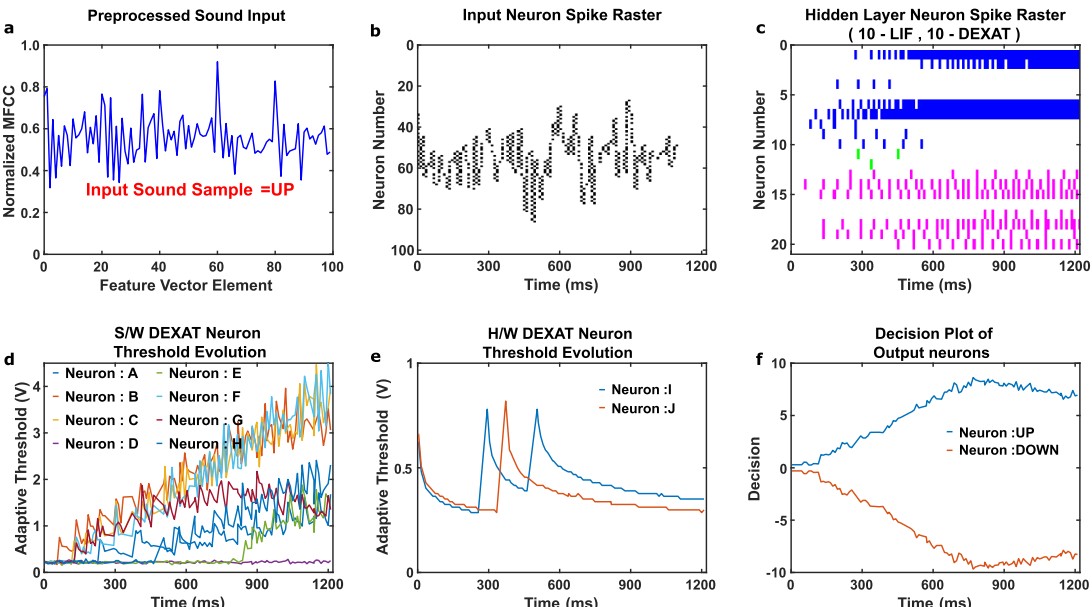

**Fig. 9 Real-time recognition of a spoken input speech sample using DEXAT based LSNN. a** Normalized and preprocessed real-time input speech sample.
**b** Input neurons spike rasters. **c** Hidden layer LIF and DEXAT neurons spike rasters. Spike rasters for neuron no. 1–10 Shown in blue are for LIF neurons,
spike rasters (11–12) in green, and (13–20) in magenta are for hardware and software DEXAT neurons respectively. **d** Adaptive thresholds of software
DEXAT neurons with variability included. **e** Adaptive thresholds of hardware DEXAT neurons. $V_{DD\_SET}$ of 3.5 V and $V_{DD\_RESET}$ of 3 V is taken in experiments.
**f** Decision output plots for the two classes evolving with time and generating correct result corresponding to input sample at the end.

in the network based on experimental data and a combined variability (C2C + D2D) parameter defined as resultant variability ($\eta_r$). We use the methodology used in[39,40] for variability analysis. In order to experimentally characterize variability, we performed both stand-alone variability (i.e., only C2C or D2D) and combined variability (i.e., C2C + D2D) experiments as shown in Supplementary Fig. 6. The resultant variability parameter ($\eta_r$) is defined such that it helps to analyze the statistical impact of variations for any generic OxRAM device (i.e., different material stacks). First, several devices are cycled multiple times for implementing a given threshold sequence. Next, for each time-step (i) the corresponding mean $\mu_i$, standard deviation ($\sigma_i$), and coefficient of variation ($\eta_i = \sigma_i/\mu_i$) of the threshold voltage values (i.e., y-axis) are calculated. Finally, median of all $\eta_i$ values is defined as the resultant variability ($\eta_r$) parameter. From Supplementary Fig. 6b, $\eta_r = 30\%$ for the Ni/HfO$_2$/Al-doped TiO$_2$/TiN stack device, for sequence S1. In order to generate the simulated neuron thresholding traces, each Y-axis value (i.e., adaptive threshold voltage) for a corresponding X-axis value (i.e., time) is drawn from a random gaussian distribution ($\mu = \mu_i$, $\sigma = \eta_r*\mu_i$). Supplementary Fig. 7 shows 10,000 simulated neuron threshold traces for different values of $\eta_r$.

### LSNN training and inference simulations

*STORE-RECALL and SMNIST.* System-level simulations for STORE and RECALL tasks and classifying SMNIST dataset are performed using a three layer recurrently connected network of spiking neurons. Hidden layer consists of LIF and adaptive neurons (ALIF/DEXAT). Our proposed DEXAT neuron model equations are integrated using Tensorflow library in python code based on[4] available on github repository. Simulations are performed in discrete time with smallest time interval "$\delta t$" taken as 1 ms. The input bits and the STORE-RECALL instructions are encoded by spiking activity of input neurons at firing frequency of 50 Hz. STORE-RECALL task is trained using the BPTT algorithm. Decision error is calculated as the ratio of number of false detected cases to total number of cases in a batch (batch size = 128). We have defined the desired minimum decision error to be <0.05. Learning rate, learning rate decay and dampening factor are taken to be 0.01, 0.8, and 0.3, respectively, in the simulations. Training and testing batch size is set to 256 for simulations. All the 60,000 images in the train set of MNIST dataset are used for training. For inference, input test image is presented from 10,000 test images in a sequential manner where each input pixel is presented in 1 time step "$\delta t$" (here 1 ms). Hence, inference of one MNIST image takes 784 time steps ($\delta t$). It is important to note that the simulation time step "$\delta t$" can be chosen to satisfy the number of time steps required for the temporal task based on the extracted values of time constants from the experimental device. Each pixel of 28 × 28 MNIST image is presented to the 80 input neurons and is encoded by the firing of the input neurons with threshold crossing method. Network architecture details are specified in Supplementary Table 6.

*Speech recognition.* This application involves initial speech data sample pre-processing step. First, an input speech sample of 1 second duration is taken from

GSC dataset at a sampling rate of 16 kHz. Next, Mel Frequency Cepstral Coefficient (MFCC) with a 30 ms window and 1 ms stride is applied to the sample. This leads to a feature vector consisting of 40 output features, extracted for all 100,503 speech samples, spanning across the 30 classes of the GSC dataset. We trained multiple LSNN networks of varying dimensions (Supplementary Fig. 15) to classify the speech data over 12 distinct classes. Here, 10 classes (yes, no, up, down, left, right, on, off, stop, and go) are taken as such from the dataset, 20 other classes of the dataset are grouped together to form an "unknown" class, while "silence" denoting absence of any speech is taken as the 12th class. For all GSC LSNN simulations (Supplementary Fig. 15) non-adaptive LIF neurons constitute 50% of the total hidden layer neurons. The remaining 50% are either DEXAT or ALIF neurons. Networks are trained using BPTT with Adam optimizer. Batch size is taken as 100. Learning rate is varied as [0.01, 0.005, 0.002, 0.001] over 10,000 training iterations [4200, 4000, 1200, 600], having dataset split of (80:10:10 i.e., train-validation-test). Post-training, inference is performed on 4890 samples to estimate the test accuracy. For full hardware end-to-end live speech recognition experiment (Fig. 9), a network (consisting of 100 neurons in input layer, 10 LIF, and 10 DEXAT neurons in hidden layer) is trained offline on a reduced 2-class GSC dataset. The two classes consist of 7711 samples each of 1 s duration, with Label "UP" (3917 samples) and Label "Down" (3794 samples), having a split (70-20-10 i.e., Train: Validation: Test). During inference, live input speech command spoken by a human is recorded in real-time using in-built microphone of the logitech C270 webcam. Preprocessing of this input sample is done in the same way as that of recorded GSC samples, by dividing it into five parts, with each part of 0.2 s duration. On each part of the sample, mean of 20 bank MFCC is extracted leading to a final feature vector with 100 features. Min–Max normalization is performed on every spoken sample. This feature vector is parsed in real-time on the frozen pre-trained LSNN realized in hardware (shown in Supplementary Fig. 10) for live classification decision.

### Data availability

Data and results presented in the plots of this study can be made available by the corresponding author, upon reasonable request. The MNIST dataset is available at https://www.tensorflow.org/datasets/catalog/mnist. The google speech commands dataset is available at https://storage.cloud.google.com/download.tensorflow.org/data/speech_commands_v0.02.tar.gz.

### Code availability

Code used in this study can be made available by the corresponding author, upon reasonable request.

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

## Acknowledgements
Authors would like to thank Prof. Tuo-Hung Hou from NCTU, Taiwan for providing $HfO_2/TiO_2$ RRAM devices. PI: M.S. would like to acknowledge partial support from: SERB-CRG/2018/001901 grant, IITD-FIRP grant and CYRAN AI Solutions. A.S. is supported by NET-JRF Fellowship of UGC. Authors would like to acknowledge the use of the LSNN code available at: https://github.com/IGITUGraz/LSNN-official. Authors would like to thank and acknowledge contribution of Sufyan Naseem Khan toward test-board fabrication and characterization.

## Author contributions
M.S. conceptualized the idea and designed the experiments. A.S. realized the DEXAT circuit. S.S.B. performed system-level simulations. All authors performed circuit–device experiments and analysis. All authors contributed to writing, discussion, and revisions.

## Competing interests
The authors declare no competing interests.
