## [Peer Review File · Nature Communications]

Reviewers' Comments:

Reviewer #1:

Remarks to the Author:

This paper proposed an adaptive threshold neuron (DEXAT) with two exponential decay elements for the threshold value after firing. The author claimed that it is advantageous compared with ALIF neuron since the long short-term memory is more controllable, and therefore the computational performance can be improved. This neuron was fabricated and characterised experimentally. The key parameters were extracted and used to form neurons in a LSNN simulation. This model was evaluated by a STORE-RECALL task and the SMNIST benchmark. The result shows that using the DEXAT neuron can improve the accuracy of LSNN in neuromorphic system. However, there are several critical issues to be addressed, especially the advantage of the proposed DEXAT neuron compared to previous implementations.

Major issues:

1. The first key point claimed in this work is that "a new dynamically varying Double EXponential Adaptive Threshold (DEXAT) neuron model for RSNNs." The model proposed in this work is a slightly improved version over the previous work (Bellec et al/ref 4). However, the author did not clearly explain why the new model has better performance. For example, why two time constants are better than a single one? What roles do the small and large time constants play in the system? Will the system performance be improved if more time constants are introduced?

2. The second key point claimed in this work is the hardware implementation of the proposed neuron model. The authors used a 6T1R circuit and discrete components to build an adaptive threshold neuron that can perform a double exponential threshold decay. Although the circuit built in this work can reproduce the behavior in the proposed model, there are still several serious problems. First of all, Table 2 is not a fair comparison, because the 6T1R circuit in this work only has the function of threshold adjustment, and it does not count the peripheral circuits (such as integrator, comparator, pulse generator, etc) that are needed to realize the complete neuron function, which is implemented in other works. In addition, the neuron designs proposed in Ref. 14 and Ref. 19 are specifically compatible with STDP, an on-chip learning scheme. From this perspective, the comparison doesn't reflect the advances of the proposed neuron. Secondly, the authors did not describe in details the reasons for using memristors to achieve threshold modulation. As can be seen from Figure 5, the authors seem to only need to implement low-pass filtering of the input signal. In order to achieve this function, is it the best choice to use a memristor? The authors also need to evaluate the performance of the proposed circuit (such as power and energy consumptions, latency, etc.) and compare them with other works. Furthermore, the authors only considered the impact of cycle-to-cycle variation on the system accuracy without further considering the impact of device-to-device variation, because the simulated system has more than one neuron units.

3. Furthermore, the actual implementation and scalability of the proposed network is also questionable. Fig. 4 shows that a single neuron requires at least an integrator (with capacitor?), a comparator, a pulse generator with 4 controllable channels, and a 6T1R circuit. Is it scalable when you need thousands of neurons in a network? What will happen to the size and power consumption?

4. For the DEXAT neuron, some operations of the OxRAM during the training and testing of the network are questionable as a neuromorphic system. For example, the $\delta t = 1\text{ms}$ was chosen throughout the experiment, which is quite long. In general, a spike-based neuromorphic system should be designed to achieve high speed processing. In line 276, the authors mentioned that δt was chosen based on the extracted value from the device. However, if a much shorter δt is applied, can the proposed neuron still retain a high accuracy? What's the limit of δt can be used? According to line 274-278, the inference of one image takes 784 ms, so going through the whole dataset once (60,000 images) would take 784 mins, which is way too long.

5. The 'STORE and RECALL task' used in this paper was not specified. The author needs to provide more details (such as the training process and the definition of decision error) in the Method section or in the Supplementary Information.

6. Also, the manuscript is not well-structured: A large proportion of the Result section looks more like 'introduction' or 'method', e.g. lines 46-65, 98-136, 154-173....

7. The network structures, such number of neurons in each module and the connections (weights) used for the two tasks should be discussed in more detail.

8. What is the advantage of using memristors to implement the DEXAT neuron while comparing to the CMOS based decay circuits? The authors should discuss this in more details (i.e. area, power consumption, operational speed, etc.).

9. Is the data of memristors in Fig. 3 all extracted from the references? Fabrication of the Ni/HfO₂/Al doped TiO₂/TiN based memristor should be presented in the method section.

10. Application of memristors in neuromorphic computing has been studied a lot (DOI: 10.1109/JPROC.2020.3004543). I suggest the authors do more literature study before they give out the conclusion in Page 5, line 52.

Minor comments :

Ref. 4 and Ref. 24 are repeating

Line 31 grammatical error: '...such as-faster'

Line 39 grammatical error: '...CBRAM) etc. there are' Also, the terms of RRAM, PCM, and CBRAM are not defined

When citing a figure, the authors used both 'Fig.' and 'Figure' in different occasions.

Line 126 grammatical errors: '...initial high resistance state (HRS) state.....' and line 127 'HRS state'

'Cycle to cycle (C2C)' is defined 5 times in the text.

Reviewer #2:

Remarks to the Author:

In this article, the authors have proposed a Double EXponential Adaptive Threshold (DEXAT) neuron model and demonstrated the algorithm with a set of oxide RAM (OxRAM) devices. The experiment shows an accuracy of 96% for the experiment with a relative variability value of 10%. In part, they built a circuit-device that can help realize the double exponential with varying time constants. Overall, I believe this is an interesting piece of study that has achieved some level of implementation of a new neuron model in non-volatile devices. However, I would recommend the authors to consider the following questions and issues.

(1) My main concern is the experiments necessary to demonstrate the benefit of the DEXAT mode is not complete. Though the authors extracted characteristic device parameters from the device measurements, there are no experiments showing the learning process. Thus, the experiment that was carried is not adequate to demonstrate the benefit of the DEXAT model. The authors obtained parameters such as the operation voltage, time, etc. from the measurements shown in figure 3. Then in Figures 5 and 6, they measured the output responses for the input spike sequence. These experiments alone could not prove the advantage of the DEXAT model. It is recommended that the authors could carry out experiments to demonstrate a full LSNN learning process in the OxRAM devices. This will be helpful in understanding how the OxRAM device variability affects the performance and compare it with the simulated values outlined in the paper.

(2) It is unclear what the circuitry looks like in the integrator and how it is designed to work with the threshold modulator circuit. It will be useful to show a simulated transient response of the proposed circuit.

Reviewer #3:

Remarks to the Author:

This manuscript reports a new silicon neuron circuit optimally designed using OxRAM device. The authors' idea to fit the neuron's adaptive characteristics to that of single OxRAM device is interesting and splendid. I believe this manuscript is worth being published provided some points are explained more clearly and mistakes are corrected.

Main points

1. In Eq. (2),

(a) p_j is precisely $1 - \delta t / \tau_a$, but expressed as $\exp(-\delta t / \tau_a)$ in line. 51. The latter approximately

equals to the former, but the former is more simple than the latter. Why $\exp(-\delta t/\tau_a)$ is used instead of $1 - \delta t/\tau_a$?

(b) $z_j(t)$ is $1/\delta t$ for 1 time step when a spike event arises and 0 otherwise. This should be explained.

2. Regarding IDLE state (Fig. 4(d)),

(a) In line 171, it is explained that the proposed circuit transits to IDLE state when the spike event concludes. But "the conclusion of the spike event" does not seem to be clearly defined. It would improve clarity if it is explicitly defined.

(b) Some readers will be interested in why 2V Vdd is used in IDLE state and 3V otherwise.

3. The authors' experimental setup is shown in Fig. S3. Information related to fabrication process of the CMOS circuit would be required.

4. In Fig. 6(b), the error bar for $\Delta 1$ is significantly larger than that for the others. Discussion on this point would extend profoundness of the manuscript.

5. In lines 212-213, it is written that Figs. 7(b) and (c) are from a random population during the learning process.

(a) In Figs. 7(b) and (c), 120 and 100 neurons are plotted respectively. So, it seems that these plots are from all the neurons.

(b) It would be more clear if what the colors (black and red) of the dots mean in Fig. 7(b).

(c) The title of Fig. 7 expresses that the plots are obtained after the training process. Which is correct?

6. In lines 232-234, the authors argue that the effect of fabrication variability can be evaluated by inserting noise. Though the authors cite references, because it is not very intuitive, explaining the reason would strongly help readers' understanding.

7. As shown in Fig. 8, performance of this work does not exceed that of LSTM. A strong driving force for studying spiking system is energy efficiency as the authors write in line. 17. Thus I believe it is imperative to discuss energy consumption of the proposed system. Power consumption of the circuit could be added in Table 2 and compared with other works.

Minor points

1. It would improve clarity of the manuscript if overall equations of the neuron and synapse models are presented in supplementary materials.

2. In line 66, reference No.24 is cited. But I could not understand how it differs from reference No.4. In addition, it seems that no explanation for the STORE and RECALL task is given in both references. The authors might mistakenly cited No.24.

3. Figure 4(a) would be more precise if a reset signal from Pulse Generator to Integrator is drawn.

4. In Fig. 8(a), it might be more clear if LSNN is explained as ALIF-based LSNN.

Takashi Kohno, M.D., Ph.D
Professor, IIS, University of Tokyo

Reviewer#1

This paper proposed an adaptive threshold neuron (DEXAT) with two exponential decay elements for the threshold value after firing. The author claimed that it is advantageous compared with ALIF neuron since the long short-term memory is more controllable, and therefore the computational performance can be improved. This neuron was fabricated and characterised experimentally. The key parameters were extracted and used to form neurons in a LSNN simulation. This model was evaluated by a STORE-RECALL task and the SMNIST benchmark. The result shows that using the DEXAT neuron can improve the accuracy of LSNN in neuromorphic system. However, there are several critical issues to be addressed, especially the advantage of the proposed DEXAT neuron compared to previous implementations.

We thank the reviewer for providing useful feedback. Below we have tried to address all concerns of the reviewer point-by-point to the best of our capacity.

MajorIssues

1. The first key point claimed in this work is that “a new dynamically varying Double EXponential Adaptive Threshold (DEXAT) neuron model for RSNNs.” The model proposed in this work is a slightly improved version over the previous work (Bellec et al/ref 4). However, the author did not clearly explain why the new model has better performance. For example, why two time constants are better than a single one? What roles do the small and large time constants play in the system? Will the system performance be improved if more time constants are introduced?

Response R1.1:

We thank the reviewer for raising this question. In the original submitted version of the manuscript we empirically demonstrated through extensive simulations that the proposed DEXAT neuron model performs better than the ALIF model in terms of: (i) convergence time (10,000 iterations faster compared to ALIF for SMNIST) and (ii) higher accuracy (2.8 % higher accuracy for SMNIST). Based on the reviewer’s remarks, below we present a detailed qualitative (assertions 1 - 4) and mathematical analysis for the reasons behind better performance of the proposed DEXAT neuron model in the revised version. The explanation has also been added in revised manuscript lines ‘106-110’ and detailed discussion is added in Supplementary Note 1 and Supplementary note 2 Supplementary material of the revised manuscript.

Detailed qualitative explanation

We use the pseudo-derivative framework equations (1)-(3) [4] to analyze the behaviour of our proposed DEXAT neuron in any generic RSNN.

$$dz_j(t) / dv_j(t) = \gamma \max\{0, 1 - |v_j(t)|\} \dots\dots\dots(1) \text{ (Pseudo-derivative)}$$

$$v_j(t) = (V_m(t) - B_j(t)) / B_j(t) \dots\dots\dots(2) \text{ (normalized membrane potential)}$$

$$\Delta E = dE / dv_j(t) = dE / dz_j(t) * dz_j / dv_j(t) \dots (3) \text{ (Hidden layer error derivative)}$$

[Here, $z_j(t)$ denotes output spike of neuron ‘j’, γ denotes the damping factor having a value less than 1, $v_j(t)$ is the normalized membrane potential, $V_m(t)$ denotes the neuron membrane potential and $B_j(t)$ is the neuron firing threshold voltage, $dE / dz_j(t)$ is the error derivative of the output layer.]

From equations (1)-(2), we observe that the pseudo-derivative is a function of the neuron's firing threshold $B_j(t)$. Thus, the adaptive behaviour of the neuron threshold determines the magnitude and variation of the

pseudo-derivative. Further, the hidden layer error gradient ΔE (equation-3) depends on the product of pseudo-derivative and error calculated at the output layer. Synaptic weight changes during training are

governed by equation (4) and are function of hidden layer error gradient ΔE . Weight updates are optimized using ADAM optimizer equations (5)-(8). Thus, synaptic weight changes are governed by the adaptive threshold decay behaviour.

$$w_t = w_{t-1} - \eta \frac{\hat{m}_t}{\sqrt{\hat{v}_t + \epsilon}} \dots \dots \dots (4)$$

$$\hat{m}_t = \frac{m_t}{(1 - a_1^t)} \dots \dots \dots (5)$$

$$\hat{v}_t = \frac{v_t}{(1 - a_2^t)} \dots \dots \dots (6)$$

$$m_t = a_1 m_{t-1} + (1 - a_1) \Delta E \dots \dots \dots (7)$$

$$v_t = a_2 v_{t-1} + (1 - a_2) \Delta E^2 \dots \dots \dots (8)$$

[\hat{m}_t is the bias-corrected first moment estimate, m_t is the 1st moment vector, v_t is the 2nd moment vector, \hat{v}_t is the bias-corrected second raw moment estimate, η is learning rate.]

Based on above set of pseudo-derivative, error gradient and weight update equations (1-8) we frame a hypothesis and some assertions to make a logical case below:

Assertion 1: Certain applications require large working memory

Sequential tasks (such as speech recognition), operating at real-time, have a large working memory requirement i.e need of several time steps for temporal processing (in revised manuscript we show 100 time steps of speech recognition, 280 time steps for SMNIST etc.). Thus to provide a long-short term memory to the network, large values of neuron adaptation time constants are necessary. Thus, in case of pure ALIF neurons, adaptation time constants equal-to or greater than the application's working memory are used for realizing the LSNN [4]. However, upon a deeper mathematical analysis (using pseudo-derivative and error gradient framework) we found that **just using a single large adaptation time constant doesn't lead to the most optimized solution and more than one time constant of specific types can lead to better outcome.**

Assertion 2: A smaller time constant assists the system in finding global minima and weight fine-tuning

From equation (3) we observe that a small pseudo-derivative value results in a small value of ΔE . Also, from equation (4) synaptic weight changes are governed by the magnitude of ΔE . A smaller time constant present in the adaptive threshold decay profile of the neuron in a LSNN can help achieve a small pseudo-derivative value (see section on mathematical explanation for proof). Thus, only with the inclusion of a small time constant in decay profile, precise fine-tuning of weight-matrix can be performed during training to achieve global error minima and thus high accuracy as also shown in Fig. R1. **Hence, we need two time constants for the neurons constituting the LSNN. Larger time constant to satisfy the working memory requirement for sequential tasks while smaller time constant to enable subtle and precise weight matrix fine-tuning.**

Assertion 3: Desired Activation Function behavior: (i) During training initially a smaller pseudo-derivative value should occur for a finite duration followed by a larger pseudo derivative value. (ii) Active neurons should experience lower value of pseudo-derivative while inactive neurons should experience higher value of pseudo-derivative for network fine-tuning and convergence.

During training, as specific neurons learn (or become sensitive to) certain specific features, their firing activity increases whenever they encounter similar features in the input stimuli. In order to consolidate learning, one would want to only minutely fine-tune (induce only small changes) the weight matrices of those neurons which have a higher firing activity (i.e. the ones which have already learnt certain feature(s)). Any drastic variation in

Figure R1. Qualitative diagram showing weight changes versus error in a DEXAT based network. During the course of training weight changes become small and aid in reaching global minima accurately. Length of the arrow denotes magnitude of weight change (Smaller arrow denotes small changes and vice versa). Network may remain stuck in a local minima state if both small and large values of pseudo-derivatives (or time constants) are not accessible to the activation functions.

the weight matrices of such neurons (which have already learnt something) may lead to loss of learning or even complete forgetting of the feature. From Assertion2, one can deduce that smaller pseudo-derivative value leads to finer weight changes and vice-versa. The proposed DEXAT model ensures that when a neuron fires frequently (i.e. smaller inter-spike interval) it automatically experiences a lower pseudo-derivative value (and consequently minute weight changes), due to our specific choice of a smaller first time constant in the DEXAT decay profile. Thus, having a faster time constant as the first time constant in DEXAT decay profile, reinforces the learning of the neuron. However, if the proposed DEXAT neuron fires infrequently (i.e. low activity, longer inter-spike time), it will encounter the second longer time constant from the decay profile (consequently inducing larger changes in its weight matrix). Further, since all the active neurons in the LSNN undergo fine tuning, a significant number of weights in the network auto converge independent of the decay of learning rate ‘ η ’ (after every ‘ n ’ iterations) in equation (4), that is conventionally done to accelerate convergence. Hence, the inherent behaviour of our proposed DEXAT neuron helps in achieving faster convergence of the network. **Thus in the proposed DEXAT neuron model the decay profile is constructed with 2 time constants such that the smaller time constant occurs first followed by the larger time constant.**

Detailed mathematical analysis and simulated proof supporting Assertions 1-3

From equation (1) and (2) we can write pseudo-derivative as

$$\frac{dz_j(t)}{dv_j(t)} = \gamma \max\left\{0, 1 - \left| \frac{V_m(t) - B_j(t)}{B_j(t)} \right| \right\} \text{-----} (9)$$

Two cases are possible w.r.t. the spike event

Case 1 : (Spike event) $V_m(t) \geq B_j(t)$

When membrane potential is greater than threshold voltage pseudo-derivative can be simplified to equation (10)

$$\frac{dz_j(t)}{dv_j(t)} = \gamma \max\left\{0, \frac{2 * B_j(t) - V_m(t)}{B_j(t)} \right\} \text{-----} (10)$$

Case 2 : (After Spike event) $V_m(t) < B_j(t)$

When membrane potential is less than threshold voltage pseudo-derivative can be simplified to equation (11)

$$\frac{dz_j(t)}{dv_j(t)} = \gamma \max\left\{0, \frac{V_m(t)}{B_j(t)}\right\} \text{ ----- (11)}$$

We analyze case 2 to see the effect of adaptation time constants and threshold decay on pseudo-derivative behaviour. After the spike event has occurred, the pseudo-derivative magnitude is governed by the ratio of membrane voltage and threshold voltage decay i.e. $\frac{dz_j(t)}{dv_j(t)} = \frac{V_m(t)}{B_j(t)}$ (assuming $\gamma=1$). Hence, larger is the value of threshold voltage $B_j(t)$, smaller is the pseudo-derivative magnitude $\frac{dz_j(t)}{dv_j(t)}$ at a time 't'. In our proposed DEXAT model there are two time constants governing the threshold decay and the threshold voltage is given by equation (12).

$$B_j(t) = b_{j0} + \beta_1 b_{j1}(t) + \beta_2 b_{j2}(t) \text{ ----- (12)}$$

$$b_{j1}(t + \delta t) = \rho_{j1} b_{j1}(t) + (1 - \rho_{j1}) z_j(t) \text{ ----- (13)}$$

$$b_{j2}(t + \delta t) = \rho_{j2} b_{j2}(t) + (1 - \rho_{j2}) z_j(t) \text{ ----- (14)}$$

[where $b_{j1}(t)$ is the term corresponding to smaller time constant as in equation (13) and $b_{j2}(t)$ is the term corresponding to larger time constant in equation (14).]

Figure R2. Pseudo-derivative behaviour comparison for DEXAT and ALIF neurons. A constant current of 50 mA is injected as input to all the three neurons, ALIF1 (with a small time constant) and ALIF2 (with a large time constant) and a DEXAT neuron (with a small and a large time constant). Neuron parameters are listed inside graph. (a) Output spikes corresponding to input current (b) Membrane potential evolution (c) Adaptive threshold behaviour and (d) Pseudo-derivative magnitude behaviour for the three neurons.

Assuming $\tau_{a1} < \tau_{a2}$, during the initial time after spike $\sim(t < 5\tau_{a1})$ threshold decay is dominated by τ_{a1} but the total threshold voltage is governed by the sum of both the decay terms i.e. $b_{j1}(t)$ and $b_{j2}(t)$ according to equation (12). This results in a larger magnitude of threshold voltage $B_j(t)$. Hence, from equation (11) the

magnitude of pseudo-derivative $\frac{dz_j(t)}{dv_j(t)}$ for initial duration ($\sim 5\tau_{a1}$) after spiking becomes small in DEXAT neuron. After the smaller time constant term $b_{j1}(t)$ decays completely for $\sim (t > 5\tau_{a1})$ and reduces to almost zero, threshold voltage value magnitude is determined by a single larger time constant term $b_{j2}(t)$. Hence, $B_j(t)$ reduces thereby increasing pseudo-derivative $\frac{dz_j(t)}{dv_j(t)}$ gradually for later part of decay. This can be observed in Fig.R2 for DEXAT neuron with two time constants $\tau_{a1} = 3 \delta t$ and $\tau_{a2} = 280 \delta t$ where δt denotes smallest time step. Fig. R2 (a) and Fig R2 (b) shows the firing events and membrane voltage evolution of the three neuron corresponding to a constant input current of 50 mA). From Fig. R2 (c), (d) we observe that a small pseudo derivative value can be realized for a finite duration after firing event using a single small value of time constant (here $3 \delta t$) in an ALIF neuron. However, in this case the threshold voltage decays rapidly and quickly reaches the baseline voltage. Hence, it cannot provide a practical working memory for temporal processing and defeats the motivation of using an adaptive neuron in RSNN for sequential tasks. On the other hand, a single large adaptation time constant (here $280 \delta t$) satisfies the working memory requirement of sequential tasks but is not able to provide the necessary small pseudo-derivative magnitude for some duration after firing event compared to DEXAT neuron as shown in Fig. R2 (d). We observe from Fig. R2 (c) that threshold voltage in our DEXAT neuron achieves a high value due to the presence of two contributing time constant terms as given in equation (7). This results in a smaller pseudo-derivative value (from equation (6)) for some duration just after spike when compared to ALIF neuron as shown in Fig. R2 (d). From Fig. R2 (c) we note that a small initial time constant also ensures that the larger threshold voltage achieved after spike decreases for some time so that pseudo derivative doesn't stay at a low value for a large duration else there are chances of the network getting stuck in local minima. Hence, DEXAT neuron helps in achieving high accuracy and faster convergence while satisfying large working memories.

Impact of dual adaptive time constants on LSNN weight matrix evolution during training

We analyze a case of training an ALIF based- versus a DEXAT based- LSNN. For this we extracted the synaptic weights during the training procedure for SMNIST using the 3 layer LSNN network as shown in revised manuscript in Fig. 1 (c). The network consists of 80 input neurons, 120 LIF and 100 DEXAT neurons in the hidden layer and 10 output neurons. We analyzed synaptic weight distribution at the start of training, after 15000 iterations and after 25000 iterations for both the networks as shown in Fig. R3 and Fig. R4. It can be observed in Fig. R3 (e)-(f) and Fig. R4 (e)-(f) that in the case of DEXAT based LSNN weight distribution do not drift much while going from 15000 training iterations to 25000 training iterations, which can be considered as fine-tuning of weight matrices (i.e. no drastic changes as accuracy improved from 94.5% to 95.7%). This happens due to the advantage of two time constants as described earlier. However, in case of ALIF based LSNN there is a significant drift in weight distribution while going from 15000 training iterations to 25000 iterations as evident in Fig. R3 (b)-(c) and Fig. R4 (b)-(c), indicating that weight-tuning was coarse and more drastic (as accuracy improved from 83.2% to 89.1%). In DEXAT based LSNN higher accuracy is achieved at the end of 25000 iterations compared to ALIF.

Figure R3. Comparison of synaptic recurrent weight matrix evolution between ALIF and DEXAT based LSNN for SMNIST application. Recurrent weights distribution for ALIF based LSNN during training: (a) Initial random, (b) After 15000 iterations, and (c) after 25000 iterations. Recurrent weights distribution in DEXAT based LSNN during training: (d) Initial random, (e) After 15000 iterations, and (f) after 25000 iterations.

Figure R4. Comparison of input-to-hidden layer weight matrix evolution between ALIF and DEXAT based LSNN for SMNIST application. Input to hidden layer weight distribution for ALIF based LSNN: (a) Initial random, (b) after 15000 iterations, and (c) after 25000 iterations. Input to hidden layer weight distribution for DEXAT based LSNN: (d) Initial random, (e) after 15000 iterations, and (f) after 25000 iterations.

Impact of using more than two exponentials ?

Based on the reviewer’s question we also investigated the impact of including more than 2 time constants in the DEXAT model. Although, by increasing the number of exponentials, one can obtain multiple different pseudo-gradients at different time intervals however the merit of a 3rd or 4th time constant was not apparent. We simulated the LSNN network for SMNIST task with neurons having 3-exponentials i.e. “TREXAT” ($\tau_{a1} = 3$, $\tau_{a2} = 280$, $\tau_{a3} = 400$, $\beta_1 = 1.21$, $\beta_2 = 12$ and $\beta_3 = 13$) and even neurons with 4-exponentials i.e. “QUADEXAT” ($\tau_{a1} = 3$, $\tau_{a2} = 280$, $\tau_{a3} = 400$, $\tau_{a4} = 500$, $\beta_1 = 1.21$, $\beta_2 = 12$, $\beta_3 = 13$ and $\beta_4 = 15$). Fig. R5 shows the comparison of test accuracies of TREXAT and QUADEXAT with our DEXAT neuron having two exponentials and an ALIF neuron with single exponential as in [4]. Accuracy was observed to be 96.3% and 95.9% after 25000 iterations for “TREXAT” and “QUADEXAT” cases respectively. Clearly the performance change after adding 3rd or 4th exponential is marginal or negligible compared to two exponentials. However realizing 3 or 4 exponentials in

hardware will be extremely challenging and complicated in place of two, thus we feel that there is no merit in going beyond 2 exponentials.

Figure R5. Test accuracy comparison on SMNIST task. Our proposed model is compared by adding one and two extra exponentials to DEXAT model to obtain triple exponentials (TREXAT) and Quadruple exponentials (QUADXAT) respectively. Adding extra exponentials beyond 2 results in negligible difference in accuracy.

2. Reviewer: *“The second key point claimed in this work is the hardware implementation of the proposed neuron model. The authors used a 6T1R circuit and discrete components to build an adaptive threshold neuron that can perform a double exponential threshold decay. Although the circuit built in this work can reproduce the behavior in the proposed model, there are still several serious problems. First of all, Table 2 is not a fair comparison, because the 6T1R circuit in this work only has the function of threshold adjustment, and it does not count the peripheral circuits (such as integrator, comparator, pulse generator, etc) that are needed to realize the complete neuron function, which is implemented in other works. In addition, the neuron designs proposed in Ref. 14 and Ref. 19 are specifically compatible with STDP, an on-chip learning scheme. From this perspective, the comparison doesn't reflect the advances of the proposed neuron.”*

Response R1.2a:

We understand the reviewer's concern related to comparison in Table 2. We would like to acknowledge an error in the original submitted version. We actually intended to compare the proposed neuron against references [21], [22], [23] and not [13], [14], [19]. We have rectified this mistake in the revised version.

1. Further, we would like to mention that the main focus and novelty of the hardware implementation part proposed in our study is the realization of the complex threshold modulation/adaptation block and not standard neuron periphery blocks such as- comparator, integrator, and pulse generator. Regarding neuron periphery blocks it's important to note that: In [14] integration is done using a membrane capacitor. In [19] authors use passive capacitor. Integrator in ref. [21] is also realised using a capacitor. Internal details of comparator and digital pulse control blocks are not provided. In [22] the integrator is realised using a capacitor and switched capacitor resistors. In [23] the integrator is realized using an OpAmp and passive elements (capacitor and resistor). Hence, we believe that implementation of peripheral blocks like integrator and comparator may be realized using any standard or hybrid (CMOS-NVM-Passive component circuits) technique involving capacitors, OpAmps etc. listed in prior art. Even though the neuron periphery blocks are not the main focus of this work, we have now implemented the full neuron functionality in end-to-end hardware RSNN demonstration (as shown in the later part of the response sheet - Fig. R9 and Fig. R10 and the revised manuscript Supplementary Fig.9 and Supplementary Fig.10).
2. It is important to note that efficient hardware realization of the adaptation behaviour is challenging as it involves (i) increasing the peak threshold value after each spike, and also (ii) multi time constant based exponential decay profile for different pseudo-derivatives. Combination of both these functions in hardware is a key novel aspect for this work. Hence, we emphasize and contrast only the threshold

adaptation block (6T-1R) with that of other neuron implementations for a fair comparison. Thus, in Table 2 we only compare the complexity of our adaptation block circuit with the corresponding adaptation circuit blocks of references [21], [22], [23].

3. Further, we could not find any prior instance in literature of full experimental demonstration of an adaptive threshold neuron for use in Recurrent Spiking Neural Networks in hardware. Hence, the comparison is performed only with existing adaptive neuron implementations i.e. [21], [22], [23] regardless of the learning rule for which they are used in the network. Thus the comparison and benchmarking presented in the revised manuscript is justified in our opinion.

Reviewer: *“Secondly, the authors did not describe in details the reasons for using memristors to achieve threshold modulation. As can be seen from Figure 5, the authors seem to only need to implement low-pass filtering of the input signal. In order to achieve this function, is it the best choice to use a memristor? The authors also need to evaluate the performance of the proposed circuit (such as power and energy consumptions, latency, etc.) and compare them with other works.”*

Response R1.2b:

1. Intrinsically occurring nonlinear conductance-modulation dynamics observed in some bi-layer OxRAM devices (detailed in Fig. 3) in revised manuscript, that we exploit in our proposed 6T-1R DEXAT circuit, is one of the key reasons for realizing the threshold modulation block with memristive devices. As explained in the text, it should be noted that not any arbitrary memristive device would lead to this functionality of the proposed 6T-1R circuit.
2. Although [21], [22], [23] present hardware neuron adaptation through modifying neuron membrane dynamics, however the adaptive function in our proposed activation (DEXAT) is significantly more complex compared to prior art (as it increases at each spike event and decays with two distinct exponential time constants).
3. In spite of implementing a simpler adaptation function (only a single exponential decay time constant for the current/voltage) [21], [22] require larger circuits and even capacitors compared to our compact 6T-1R realization. Specifically, in [21], the adaptation circuit consists of a capacitor, a tunable resistor composed of 7 transistors and other blocks like multiplexers, buffers etc. In [22] the adaptation circuit is built using a mixed signal switched capacitor circuit consisting of five capacitors and more than 10 transistors. Our proposed circuit is significantly simpler using only 1 OxRAM device and 6 CMOS transistors. Other works in [a], [b], [c] also involve use of capacitors and multiple CMOS transistors to realize adaptation blocks. In [a] four MOS transistors and a capacitor is used. In [b] six MOS transistors and a capacitor is used. In [c] five transistors and one capacitor is used. Thus, the illustrated circuit efficiency and unique device property exploitation is the primary motivation behind using special memristor based circuits.

Apart from direct functionality there are multiple auxiliary benefits of building such circuits using memristors, that we'd like to point out such as:

4. OxRAM devices can be integrated in the back-end of the line (BEOL), 3D integrated saving precious silicon area.
5. OxRAM devices are CMOS compatible and offer non-volatility that can be exploited for other purposes in the network such as synaptic weight storage etc.
6. It is likely that future hardware neural networks include dense memristive synaptic arrays. If memristive devices are already present on the silicon die/fab-process used for synaptic application then building efficient neuron/activation circuits (like the proposed DEXAT) using the same devices will be an added advantage for the overall system functionality and efficiency.

Further based on the reviewer's suggestion we also investigated the possibility of using pure CMOS (non-memristor based) circuits for the realization of the proposed DEXAT behavior. Following observation was made:

- We note that in literature, circuits such as log domain low pass filter (LPF), Differential pair integrator (DPI) are used as functional blocks to emulate membrane dynamics of neuron circuits and also to realize exponential decaying postsynaptic currents in synapses [d], [e]. In such circuits, transistors are used in subthreshold mode and the input spike voltage signal is integrated (i.e. low pass filtered) to produce an output current with exponential rise and decay. However, in our proposed model we desire a voltage decay behaviour with a double exponential decay of different time-scales. Further, all these circuits used for low pass filtering utilize a capacitor that adds to the silicon area. Low pass filtering of a square input spike pulse using a passive RC filter would be area inefficient and result only in a single rise/decay time constant. Moreover peak adaptation will still be missing.

In order to comprehensively address the reviewer's concern related to choice of exact method of hardware realization of the proposed DEXAT neuron we also simulated digital synthesized ASIC blocks (on multiple technology nodes) and FPGA implementation of our DEXAT neuron's threshold modulation block. We performed detailed power, energy and area analysis of these implementations and compared them with the closest analog adaptive neuron circuits from literature. Response R3.7 presents the details of this extensive benchmarking. Based on the analysis we are convinced with the merit of using memristive circuits for realizing the threshold modulator block (6T-1R CMOS-OxRAM circuit), owing to its very high area efficiency and energy efficiency (at scaled nodes ~ approaching 28 nm and below) for the proposed DEXAT neuron.

References for R.1.2b

- [a] Arthur J V., Boahen KA. Synchrony in silicon: The gamma rhythm. IEEE Trans Neural Networks. 2007;18(6):1815–25.
- [b] Livi P, Indiveri G. A current-mode conductance-based silicon neuron for Address-Event neuromorphic systems. Proc - IEEE Int Symp Circuits Syst. 2009;2898–901.
- [c] JH. Wijekoon and P. Dudek. "Compact silicon neuron circuit with spiking and bursting behaviour", Neural Networks 21.2-3: 524-534.
- [d] Indiveri G, Linares-Barranco B, Hamilton TJ, van Schaik A, Etienne-Cummings R, Delbruck T, et al. Neuromorphic silicon neuron circuits. Front Neurosci. 2011
- [e] Bartolozzi C, Indiveri G. Synaptic dynamics in analog VLSI. Neural Comput. 2007;19(10):2581–603.

Reviewer: "Furthermore, the authors only considered the impact of cycle-to-cycle variation on the system accuracy without further considering the impact of device-to-device variation, because the simulated system has more than one neuron units."

Response R1.2c:

We regret the confusion caused, however we would like to point out that as suggested by the reviewer all network results included in the revised manuscript contain both cycle-to-cycle (C2C) and device-to-device (D2D) variability. In our full network analysis we have incorporated variability on the adaptive threshold voltage of each neuron in the network based on experimental data and a new combined variability parameter (C2C+D2D) defined as **resultant variability (η_r)**. In order to experimentally characterize variability, we performed both stand-alone variability (i.e. only C2C or D2D) and combined variability (i.e. C2C+D2D) experiments as shown in Supplementary Fig. 6 of revised manuscript. The resultant variability parameter (η_r) is defined such that it helps to analyze the statistical impact of variations for any generic OxRAM device (i.e. different material stack).

First, several devices are cycled multiple times for implementing a given threshold sequence. Next, for each time-step (i) the corresponding- mean (μ_i), standard deviation (σ_i), and coefficient of variation ($\eta_i = \sigma_i/\mu_i$) of the threshold voltage values (i.e. y-axis) are calculated. Finally, median of all η_i values is defined as the resultant variability (η_r) parameter. In Supplementary Fig. 6b, for sequence S1, $\eta_r = 30\%$ for the Ni/HfO₂/Al doped TiO₂/TiN device. Using the experimentally extracted value of η_r and DEXAT model equations we generate multiple traces of neuron thresholding behavior which capture both D2D and C2C effects inside the simulated network (Fig. R6). In order to generate the simulated neuron thresholding traces, each Y-axis value for a corresponding X is drawn from a random gaussian distribution (whose $\mu = \mu_i$, $\sigma = \eta_r * \mu_i$). Different devices/material stacks will lead to different values of η_r , thus we analyzed the network behavior over a wide

range of η_r values (0 % to 40 %) as shown in Fig. R6, and also in revised manuscript Fig. 8(c). Even for the extreme case of $\eta_r = 40\%$, the accuracy drop was found to be only $\sim 2.1\%$ in case of SMINIST application thus proving the network and neuron's robustness towards both C2C+D2D variability.

Figure R6. Simulated DEXAT neuron adaptive threshold cycles using resultant variability parameter. Each curve shows 10,000 simulated neuron traces for sequence S1 capturing effect of both C2C+D2D variability. 10,000 traces are representative of 100 neurons for 100 cycles (or X neurons for 10,000 / X cycles). (a) $\eta_r = 10\%$, (b) $\eta_r = 30\%$ and (c) $\eta_r = 40\%$.

3. Furthermore, the actual implementation and scalability of the proposed network is also questionable. Fig. 4 shows that a single neuron requires at least an integrator (with capacitor?), a comparator, a pulse generator with 4 controllable channels, and a 6T1R circuit. Is it scalable when you need thousands of neurons in a network? What will happen to the size and power consumption?

Response R1.3:

We understand the reviewer's concern, however we would like to respectfully disagree with the reviewer regarding the point on scalability.

1. We would like to point out that for almost all hardware implementations of neuron designs in literature, the neuron periphery blocks (integrator, pulse-generator, comparator) are required and it is not just limited to our case. In literature, integrator circuits and adaptive threshold circuits consist of even large membrane capacitors as explained earlier in R.1.2 [21], [22], [a], [b], [c]. In spite of these overheads there have been several large scale hardware demonstrations consisting of 1000s of neurons in literature [f], [g]. Further, if the neuron is an adaptive neuron it may require even more transistors, passives and complex circuits for the realizing the threshold adaptation blocks as pointed out in R.1.2, thereby making the system less area efficient and less scalable. We have clearly outlined the area efficiency of our adaptation block circuit compared to literature in R.1.2, thus we feel that the proposed threshold modulator block (which requires no capacitors) based on bi-layer OxRAM device is clearly more practical and scalable compared to other related ideas presented in literature.
2. Although our pulse generator block generates four control signals, two of these signals are inverted forms of the other two signals and can be obtained by using CMOS inverters. Hence, the pulse generator effectively has two channels. Further, the output control signals are a pair of simple square pulses unlike complex pulse shapes as used in some literature implementations [h]. Such pulses can be realized using digital blocks.

References for R.1.3

(f) Ishii, M., et al. "On-Chip Trainable 1.4 M 6T2R PCM Synaptic Array with 1.6 K Stochastic LIF Neurons for Spiking RBM." 2019 IEEE International Electron Devices Meeting (IEDM). IEEE, 2019.

(g) Chen, Gregory K., et al. "A 4096-neuron 1M-synapse 3.8-pJ/SOP spiking neural network with on-chip STDP learning and sparse weights in 10-nm FinFET CMOS." IEEE Journal of Solid-State Circuits 54.4 (2018): 992-1002.

(h) Xinyu Wu, 'Homogeneous Spiking Neuromorphic System for Real World Pattern Recognition', IEEE JETCAS.

4. For the DEXAT neuron, some operations of the OxRAM during the training and testing of the network are questionable as a neuromorphic system. For example, the $\delta t = 1\text{ms}$ was chosen throughout the experiment, which is quite long. In general, a spike-based neuromorphic system should be designed to achieve high speed processing. In line 276, the authors mentioned that δt was chosen based on the extracted value from the device. However, if a much shorter δt is applied, can the proposed neuron still retain a high accuracy? What's the limit of δt can be used? According to line 274-278, the inference of one image takes 784 ms, so going through the whole dataset once (60,000 images) would take 784 mins, which is way too long.

Response R1.4:

Regarding timescales

The reviewer has raised a very interesting point. Please note that the HfO₂/TiO₂ OxRAM device has switching speed in order of \sim milliseconds, thus we choose a $\delta t = 1\text{ms}$ for these devices. However, other similar devices which have a faster switching time (\sim microseconds, shown in Fig. 3 (b),(d) in revised manuscript) would lead to a much lower value of δt . We observe that in literature, broadly two approaches to neuromorphic computing exist; (i) First approach aims at realizing bio-mimetic or biologically realistic time constants (i.e. order of milliseconds to seconds) as reported in [i]. Such neuromorphic circuits may be more suitable for processing real-world sensory stimuli that occur at biological timescales; example- tasks such as spoken speech/voice/sound processing etc. (ii) Second approach targets building highly accelerated bio-inspired

neuromorphic systems (i.e. time constants 1000 X greater than biology), for example in [j]. The reviewer is correct in pointing out that MNIST may not be the most well-suited application to illustrate the benefit of our network's timescale. Thus in the revised manuscript **we have now demonstrated live spoken voice command recognition based on google speech dataset.** The stimuli of human spoken voice commands work at more biological timescales so these were efficiently processed in real time using the proposed DEXAT neurons based RSNN as detailed in Fig. R9 and Fig. R10.

On Accuracy vs Timescales

To address the reviewer's question, we convert the iterations on X-axis in absolute time scale and plot the Accuracy vs Time graphs as shown in Fig. R7. We observe that a Pt/PCMO/TiN/Pt device which has switching speed in microseconds with $\delta t = 1\mu\text{s}$ as extracted in Fig. 3(b),(f) in revised manuscript achieves a high accuracy in much lesser time compared to the Pt/PCMO/N-doped TiN/Pt. In such case (with faster devices), it would be possible to do the inference for SMNIST task in 784 μs , and for an epoch it would take less than a minute (0.784 minutes). So, in one way the limit on δt is imposed by the switching speed of the device. Hence, the accuracy will not be affected with a smaller δt .

Figure R7. Learning accuracy graphs with absolute time on X-axis. (a) A device with a lower switching time (\sim ms) **(b)** A device with a faster switching time (\sim μs) can significantly reduce the real-world physical training time consumed over training epochs.

Further, in the revised manuscript we have also realized the DEXAT neuron threshold modulator block using digital FPGA and simulated digital ASIC hardware blocks (see Supplementary note 3 and Supplementary note 4). For the digital implementations as there is no physical limitation of the OxRAM device switching speed, a more aggressive $\delta t \sim 84 \text{ ns}$ was used.

References for R.1.4

[i] Chicca E, Stefanini F, Bartolozzi C, Indiveri G. Neuromorphic electronic circuits for building autonomous cognitive systems. Proc IEEE. IEEE; 2014;102(9):1367–88.

[j] T.Wunderlich,A.F.Kungl,E.M Müller,J.Schemmel,and M. Petrovici, “Brain-inspired hardware for artificial intelligence:Accelerated learning in a physical-model spiking neural network,”Lecture Notes in Computer Science, p. 119–122, 2019. [Online].Available: <http://dx.doi.org/10.1007/978-3-030-30487-410>

5. The ‘STORE and RECALL task’ used in this paper was not specified. The author needs to provide more details (such as the training process and the definition of decision error) in the Method section or in the Supplementary Information.

Response R1.5:

We take note of the reviewer's comments and have added lines ‘71-75’ in revised manuscript to provide additional background on the STORE and RECALL task .

‘Authors in [k] have demonstrated the benefits of an ALIF based LSNN using a STORE-RECALL task. STORE and RECALL is a delayed response task that tests the capability of a network for possessing ‘short-term’ memory or a working memory. Working memory is defined as the ability to store and manipulate information over a short duration of time and forms the basis of temporal and cognitive processing [l].’

Also as per the suggestion of the reviewer we have now added the required details about STORE and RECALL tasks in ‘Simulations’ section of Methods in lines ‘354-358’. A new proper reference [k] is also cited now in revised manuscript for this task.

‘The input bits and the STORE-RECALL instructions are encoded by spiking activity of input neurons at firing frequency of 50Hz. STORE-RECALL task is trained using the BPTT algorithm. Decision error is calculated as the ratio of number of false detected cases to total number of cases in a batch (batch size = 128). We have defined the desired minimum decision error to be less than 0.05.’

References for R.1.5

[k] Bellec G, Scherr F, Hajek E, Salaj D, Legenstein R, Maass W. Biologically inspired alternatives to backpropagation through time for learning in recurrent neural nets. arXiv. 2019;1–37.

[l] M. T. Todd, Y. Niv, and J. D. Cohen, “Learning to use working memory in partially observable environments through dopaminergic reinforcement,” Adv. Neural Inf. Process. Syst. 21 - Proc. 2008 Conf., pp. 1689–1696, 2009.

6. Also, the manuscript is not well-structured: A large proportion of the Result section looks more like ‘introduction’ or ‘method’, e.g. lines 46-65, 98-136, 154-173....

Response R1.6:

We take note of this and have now modified the structure of the revised manuscript.

7. The network structures, such number of neurons in each module and the connections (weights) used for the two tasks should be discussed in more detail.

Response R1.7:

Details of the 3 layer recurrent network for both tasks are included in the Methods section. We have now added the network architecture details in the Supplementary Table 4 and also copied the same below for convenience of the reviewer.

SMNIST Architecture			STORE-RECALL Architecture	
Network Layer	No. of neurons	Synaptic Weights	No. of neurons	Synaptic Weights
Input Layer	80	(80 x 220) = 17,600	100	(100x20) = 2000
Hidden layer	120 LIF + 100 DEXAT	(220x220) = 48,400	10 LIF + 10 DEXAT	(20x20) = 400
Output Layer	10	(220x10) = 2,200	10	(20x10) = 200

8. What is the advantage of using memristors to implement the DEXAT neuron while comparing to the CMOS based decay circuits? The authors should discuss this in more details (i.e. area, power consumption, operational speed, etc.).

Response R1.8: We have partially addressed this concern in detail in responses R.1.2 and R.1.3. Here we provide few more justification; In literature, **single-exponential** rise/fall decays have been realized using CMOS current domain low pass filters (h)-(i). In all such circuits a passive capacitor is used which results in significant area penalty. Even a single exponential decay CMOS circuit as shown in Fig. R8 (for ADeX neuron) requires at least 9 transistors. Our proposed activation function requires a more complicated double exponential decay behavior.

Figure R8. Exponential current generating circuit used in AdEX neuron implementation of [22].

Further, we would also like to point out that if a neuromorphic system has to interact with a biological time-scale stimuli like speech/sound processing, the time constants need to be on the order of a few 100's of milliseconds [i] going up to a few seconds. Realizing such large time constants through on-chip capacitors for each activation function would be impractical. Thus the proposed 6T1R circuit is an optimized design for the purpose. We kindly request the reviewers to read responses R.1.2, R.1.3 and R.1.8 in unison as they comprehensively address the primary question related to merit of using a memristive circuit for realizing the DEXAT.

9. Is the data of memristors in Fig.3 all extracted from the references? Fabrication of the Ni/HfO2/Al doped TiO2/TiN based memristor should be presented in the method section.

Response R1.9:

Data for Fig.3 (a), (b), (c) is extracted from references. As per reviewer suggestion, fabrication details of Ni/HfO₂/Al doped TiO₂/TiN have now been added in the methods section of the revised manuscript paper.

10. Application of memristors in neuromorphic computing has been studied a lot (DOI: 10.1109/JPROC.2020.3004543). I suggest the authors do more literature study before they give out the conclusion in Page 5, line 52.

Response R1.10:

As per the reviewer's concern, we have removed this statement from the revised manuscript.

Minor comments :

a) Ref.4 and Ref. 24 are repeating

We have corrected this error and added the proper reference now as ref [26] in the revised manuscript in place of Ref [24].

b) Line 31 grammatical error: '...such as-faster'

Corrected.

c) Line 39 grammatical error: '...CBRAM) etc. there are' Also, the terms of RRAM, PCM, and CBRAM are not defined

Added.

d) When citing a figure, the authors used both 'Fig.' and 'Figure' in different occasions.

Line 126 grammatical errors: '...initial high resistance state (HRS) state.....' and line 127 'HRS state'

Corrected.

e) 'Cycle to cycle (C2C)' is defined 5 times in the text.

Corrected.

Reviewer #2 (Remarks to the Author):

In this article, the authors have proposed a Double EXponential Adaptive Threshold (DEXAT) neuron model and demonstrated the algorithm with a set of oxide RAM (OxRAM) devices. The experiment shows an accuracy of 96% for the experiment with a relative variability value of 10%. In part, they built a circuit-device that can help realize the double exponential with varying time constants. Overall, I believe this is an interesting piece of study that has achieved some level of implantation of a new neuron model in non-volatile devices. However, I would recommend the authors to consider the following questions and issues.

We thank the reviewer for appreciating our study and providing useful comments which have helped us to further enhance the manuscript.

(1) My main concern is the experiments necessary to demonstrate the benefit of the DEXAT mode is not complete. Though the authors extracted characteristic device parameters from the device measurements, there are no experiments showing the learning process. Thus, the experiment that was carried is not adequate to demonstrate the benefit of the DEXAT model. The authors obtained parameters such as the operation voltage, time, etc. from the measurements shown in figure 3. Then in Figures 5 and 6, they measured the output responses for the input spike sequence. These experiments alone could not prove the advantage of the DEXAT model. It is recommended that the authors could carry out experiments to demonstrate a full LSNN learning process in the OxRAM devices. This will be helpful in understanding how the OxRAM device variability affects the performance and compare it with the simulated values outlined in the paper.

Response R2.1:

We thank the reviewer for this useful suggestion to improve the manuscript. We would like to point out the following additions to the reviewers:

1. We have now included a detailed qualitative and mathematical justification behind the better performance achieved in training due to the proposed DEXAT activation function in R.1.1. Irrespective of the type of hardware realization for DEXAT, in R.1.1. we provide a theoretical justification behind the merits of the proposed neuron model.
2. Based on the reviewer's suggestion we took up the challenging task of demonstrating the full end-to-end LSNN with hardware CMOS-OxRAM DEXAT neurons integrated in real-time. In order to complete these experiments we had to design and fabricate a new experimental test-bench (two additional PCBs) as shown in Fig. R10. Further in order to illustrate the utility of the concept we choose real-world applications like recognition of spoken voice commands (based on google speech dataset). Design flow of the new end-to-end LSNN experiment is shown in Fig. R9. **These experiments illustrated that (i) 6T1R hardware CMOS-OxRAM DEXAT neuron threshold modulator circuit functions as conceptualized when used inside a full LSNN and (ii) LSNN is resilient to OxRAM device variability in real-time use cases.**

Movie of experimental demonstration of live speech recognition using proposed OxRAM-based hardware DEXAT neurons can be viewed at the following link:

<https://drive.google.com/file/d/16-rHAegdOitPq4F0IIbNTh5nOzZOLCo8/view?usp=sharing>

Full LSNN Experimental Test Bench

The new end-to-end experimental setup is designed to be highly flexible and reconfigurable. It allows virtualization/partitioning of the network blocks (neurons, synapses, learning-rules) between hardware and software as per availability of resources (devices, arrays, packaging) and total network size. In our experiments we realized LIF neurons and synapses in software and partitioned DEXAT neurons in both hardware and software in order to emulate multiple situations. The threshold of hardware DEXAT neuron adapts based on the input stimuli. Adaptive threshold levels from hardware DEXAT circuits are continuously sampled and fed in real-time to the rest of the network by an on-board ADC. Thus, real experimental D2D and C2C variability of OxRAM devices is reflected inside the LSNN network. In order to achieve this we had to include a dedicated

microcontroller, ADC, DAC and switches on the first custom fabricated ‘parent-PCB’ shown in Fig. R10. A second daughter-PCB (see Fig. R10) consisting of DEXAT threshold modulator circuit components was fabricated to interface with the parent-PCB. Both PCBs together along with the host computer help to realize the hybrid LSNN as shown in Fig. R10 and the video above.

Figure R9. Block schematic of experimental setup. The setup is used for demonstrating LSNN end-to-end speech recognition tasks using real hardware CMOS-OxRAM DEXAT neuron adaptation blocks.

Description of speech recognition experiments

The task performed was classification of spoken voice command classes (Yes, No, Up, Down, Right, Left, On, Off, Go, Stop) from Google speech dataset. We use two classes ‘Up’ and ‘Down’ from the master dataset.

Preprocessing

A speech sample of 1 second is taken at a sampling rate of 16 kHz and is divided into 5 parts, each of 0.2 seconds. For each 0.2 seconds sample, mean of 20 bank MFCC is extracted. Hence, we obtain a feature vector consisting of 100 features for the given speech sample. Min-Max normalization is performed over all data. After this pre-processing, speech data is fed to the network for recognition.

Network Realization

For this experiment, all LIF neurons are realized in software and total DEXAT neurons are divided between hardware and software as shown in Fig. R9. Communication between the software defined neurons and real hardware neuron circuits is synchronized with the help of controller and host-computer. (Standard peripheral neuron blocks like integrator and comparator are implemented in software, while the most crucial neuron block i.e. the DEXAT adaptation block is realized in real CMOS-OxRAM circuit). The network (100 neurons in input layer, 10 ALIF - 10 DEXAT neurons in Hidden layer) is trained using BPTT over 7711 samples each of 1 second with Label: UP (3917 samples) and Label: Down (3794 samples), with train split (70-20-10 i.e. Train: Validation:Test) in software for this task. For inference, speech samples are preprocessed and parsed to the network. The inference takes place temporally over time as each input feature is presented in time step ‘ δt ’ where ‘ δt ’ = 10 ms. During the real-time inference, a serial input is synchronously fed to the PCBs whenever a hardware DEXAT neuron in the network spikes. As a result, the control programming signals (SET/RESET) are applied to the hardware DEXAT neuron(s). The real-time hardware adaptive thresholds are transmitted to the ADC on board and is provided to the network interface code in real-time. The hybrid hardware-software network works in unison and in real-time to infer the input speech sample. Network partitioning between software and

hardware neurons was done for practical reasons such as large PCB boards and fewer devices, smaller array size. Since we were limited to design only 2-layer PCBs for this study the PCB size grew fast with provisioning interfacing for additional hardware neurons. Thus the total number of hardware neurons was kept low (< 5). However a 4-layer or higher layer PCB would further lead to a large-scale hardware demo.

Fig. R10. End to End experimental setup. Hardware experimental setup showing fabricated PCBs with different blocks, used for demonstrating LSNN hardware inference on live speech recognition tasks.

End to End hardware experiments:

A 3 layer LSNN as shown in revised manuscript Fig. 1 (c) is used with 10 LIF and 10 DEXAT neurons in the hidden layer for below described experiments out of which 8 are software DEXAT neurons and 2 are hardware DEXAT neurons. For the hardware DEXAT neurons OxRAM device variabilities (C2C+D2D) are inherently captured in the network in real-time. Time step δt is taken as 10 ms. Inference is performed for live speech samples spoken in real time. Fig. R11 shows the result for real-time spoken speech-sample. The pre-processed real-time spoken speech sample 'UP' is shown in Fig. R11 (a). This pre-processed sample is then given to the network based on which spiking in the input neurons occurs as shown in Fig. R11 (b). Fig. R11 (c) shows the spike rasters for the hidden layer neurons during the speech recognition process. Fig. R11 (d) and Fig. R11(e) shows the adaptive thresholds of the DEXAT neurons defined in software and hardware respectively during the inference. Fig. R11 (f) shows the decision generated at the output layer with time. At the end of the inference the output decision value corresponding to 'UP' is higher than that corresponding to 'DOWN' signifying a correct decision. Apart from the real-time speech sample, we performed multiple cycles of experiments on pre-recorded input speech samples from the Google speech dataset corresponding to the two classes 'UP' and 'DOWN'. An example is shown in Fig. R12. We have added the results of end to end real-time experiments in the revised manuscript lines '254-268' of the revised manuscript. Also added the details of experimental setup and pre-processing of speech data in Methods section in lines '298-321'. We have also added Supplementary Figure. 9 and Supplementary Fig. 10 showing our setup design flow and fabricated experimental setup. Also, multiple cycle results on google speech dataset samples are added in Supplementary Fig. 11 (a)-(f) of revised manuscript.

Figure R11. Speech recognition result for DEXAT based LSNN network on a real time spoken speech sample. Two hardware DEXAT neurons are used while eight DEXAT neurons are realized in software inclusive of D2D+C2C variability. (a) Normalized and pre-processed real-time input speech sample. (b) Input neurons spike rasters. (c) Hidden layer LIF and DEXAT neurons spike rasters. Spike rasters for neuron no. 1 to 10 shown in blue are for LIF neurons, spike rasters (11 to 12) in green and (13 to 20) in magenta are for hardware and software DEXAT neurons respectively. (d) Adaptive thresholds of software DEXAT neurons with both C2C and D2D variability included. (e) Adaptive thresholds of hardware DEXAT neurons. V_{DD_SET} of 3.5 V and V_{DD_RESET} of 3 V is taken in experiments. (f) Decision output plots for the two classes evolving with time and generating correct result corresponding to input sample at the end.

Figure R12. Speech recognition result for DEXAT based LSNN network on a speech sample taken from Google speech dataset. Two hardware DEXAT neurons are used while eight DEXAT neurons are realized in software inclusive of D2D+C2C variability. (a) Normalized and pre-processed real-time input speech sample. (b) Input neurons spike rasters. (c) Hidden layer LIF and DEXAT neurons spike rasters. Spike rasters for neuron no. 1 to 10 shown in blue are for LIF neurons, spike rasters (11 to 12) in green and (13 to 20) in magenta are for hardware and software DEXAT neurons respectively. (d) Adaptive thresholds of software DEXAT neurons with both C2C and D2D variability included. (e) Adaptive thresholds of hardware DEXAT neurons. (f) Decision output plots for the two classes evolving with time and generating correct result corresponding to input sample at the end.

(2) It is unclear what the circuitry looks like in the integrator and how it is designed to work with the threshold modulator circuit. It will be useful to show a simulated transient response of the proposed circuit.

Response R2.2:

We thank the reviewer for the suggestion.

1. We have partially addressed this concern also in response R.1.2 of this response sheet. We would like to mention that in terms of hardware realization, we mainly want to focus on the complex threshold adaptation block of the proposed DEXAT. The regular neuron periphery blocks (integrator, comparator) can be used from state-of-the-art literature. In R.1.2 we have detailed that even for simple adaptive neurons hardware realization of the adaptation block is not trivial and is very complex.
2. We analyzed the implementation of peripheral blocks used in various neuron implementations in literature. For example, in [14], integration is done using a membrane capacitor. In [19] authors use a Phase Change Memory (PCM) device and passive components like capacitor for integrator block. The integrator in ref. [21] is realised using a capacitor. Internal details of comparator and digital pulse control blocks are not disclosed. In [22] integrator is realized using a capacitor and switched capacitor resistors. In [23] the integrator is realized using an OpAmp and passive elements (capacitor and resistor). Hence, we find that the implementation of peripheral blocks like integrator and comparator is done conventionally by using components like capacitors, OpAmps or a mix of emerging memory devices and is mostly similar in all neuron designs.
3. However, the adaptation behaviour is realized using different circuit designs. Hence, we emphasize the working of this important block and contrast this block with that in other neuron implementations.
4. However, considering the reviewer's comment for the need of proving full functionality we have realized the peripheral circuit blocks of DEXAT neuron, like integrator and comparator using a mix of software, ADC and DAC as shown in Fig. R9 and performed full end to end experiments using the setup shown in Fig. R10 in our revised manuscript for speech recognition task.
5. Further for the sake of a strong proof we have also realized an all digital DEXAT neuron on an FPGA as detailed in Supplementary note 3.

Reviewer #3 (Remarks to the Author):

This manuscript reports a new silicon neuron circuit optimally designed using OxRAM device. The authors' idea to fit the neuron's adaptive characteristics to that of single OxRAM device is interesting and splendid. I believe this manuscript is worth being published provided some points are explained more clearly and mistakes are corrected.

We thank the reviewer for appreciating the work and providing useful comments for improving the manuscript. We have added responses to each individual comment.

MainPoints

1. In Eq. (2),

(a) ρ_j is precisely $1 - \delta t/\tau_a$, but expressed as $\exp(-\delta t/\tau_a)$ in line. 51. The latter approximately equals to the former, but the former is more simple than the latter. Why $\exp(-\delta t/\tau_a)$ is used instead of $1 - \delta t/\tau_a$?

(b) $z_j(t)$ is $1/\delta t$ for 1 time step when a spike event arises and 0 otherwise. This should be explained.

Response R3.1:

a) We note that the expansion of exponential decay term (equation (15)) upto only two expansion terms ($1 - \delta t/\tau_a$) is accurate when $\delta t/\tau_a$ is small i.e. when τ_a is large. However, for small values of τ_a ($1 < \tau_a < 10$) there is a significant difference in magnitude of voltage decay for approximated expansion when compared to full exponential as can be seen from Fig. R13 (a). Fig 13 (b) plots the error in magnitude resulting due to approximation with time constant values. In our DEXAT model the total decay is governed by the sum of the decay components from the smaller and the larger time constants terms. Hence, to get an accurate value we used the $\exp(-\delta t/\tau_{a1})$, $\exp(-\delta t/\tau_{a2})$ terms as ρ_{j1} and ρ_{j2} . However, we believe the approximated expression as suggested by the reviewer can also be used if time constants fall in a higher range.

$$\exp(-\delta t/\tau_a) = 1 - \delta t/\tau_a + \delta t/\tau_a^2 + \delta t/\tau_a^3 + \dots (15)$$

Figure R13. Effect of approximating the exponential function with increasing time constant. (a) Plotted magnitude of exponential function ($\exp(-\delta t/\tau_a)$) without approximation and with approximation to two terms (i.e. $1 - \delta t/\tau_a$) for different time constants (τ_a) (b) Plotted error magnitude between full ($\exp(-\delta t/\tau_a)$) and approximated ($1 - \delta t/\tau_a$) exponential expression for different time constants.

Further, taking into account the reviewer's suggestion we tried a fully digital implementation of our DEXAT threshold modulator function on a FPGA where approximation becomes a necessity and have added details in Supplementary note 3 in revised manuscript. We note in this case that we have to use an approximation of exponential expansion as the reviewer has pointed to write a synthesizable code in Verilog. Also, based on the available precision and complexity needed one can decide on the number of terms to be included in the expansion. For example, for a 16 bit fixed point precision we could approximate the exponential to three terms of equation (15). For a 8 bit fixed point precision, we could approximate to two terms of equation (15). Fig. R14 shows the DEXAT adaptive behaviour simulated and synthesized for FPGA implementation in

Verilog for 16 bit fixed point precision. Table R1 shows that for realizing increased precision DEXAT, number of resources on FPGA increase drastically. With the limited precision of 8-bit fixed point (i.e. 4 bit integer, 4 bit fraction) any value of τ_a upto 32 dt can be realized. When the precision is increased to 16 bit fixed point (i.e. 8 bit integer, 8 bit fraction) any value of τ_a upto 512 dt can be realized. Adding each bit in the decimal part of fixed point can double the achievable τ_a limit. The resolution can be chosen according to the requirement of setup but an increased resolution comes at the cost of additional FPGA resources. With highest precision and no limitation on τ_a (using IEEE754 standard floating point), the hardware usage goes up by multiple folds compared to fixed point setup as shown in Table R1. From Table R1 it is clear that even for an implementation with limited 8 bit fixed point precision 79 LUTs and 41 registers are required.

Figure R14. Digital implementation of DEXAT neuron behaviour. DEXAT adaptive threshold behaviour obtained using 16 bit precision on Xilinx Artix 7 FPGA.

Table R1: FPGA implementation of DEXAT neuron block.

Number system	Total Bits	Decimal Bits	Tau Limit (dt)	LUTs	Reg	DSP
Fixed Point	8	4	32	79	41	0
Fixed Point	16	8	512	22	33	6
Floating Point	32	IEEE754	No limit	3091	1361	12

b) ' $z_j(t)$ ' denotes the output spike of the neuron. As per the neuron model there is an increase in the threshold voltage at each spike event. We have written an equation of DEXAT model for reference here in equation (16). From this equation we note that increase in the threshold voltage is governed by the second term $(1-\rho_j) z_j(t)$. The incremental value gets added to the threshold voltage only when an output spike occurs and hence $z_j(t)$ only becomes high when the neuron fires. Otherwise, when the neuron threshold voltage is decaying this term should be zero and hence $z_j(t)$ is zero. During this time period only decay of threshold voltage takes place by multiplying the previous value of threshold by the exponential decay term ρ_j .

$$b_j(t + \delta t) = \rho_j b_j(t) + (1 - \rho_j) z_j(t) \dots\dots\dots(16)$$

2. Regarding IDLE state (Fig. 4(d)),

(a) In line 171, it is explained that the proposed circuit transits to IDLE state when the spike event concludes. But "the conclusion of the spike event" does not seem to be clearly defined. It would improve clarity if it is explicitly defined.

Response R3.2:

We thank the reviewer for pointing out this ambiguity in text. The spike occurs as soon as the comparator output goes high following which control signals are generated by the pulse generator. The circuit actually goes into

IDLE state when the adaptive threshold reaches the baseline voltage due to continuous application of RESET pulse in the absence of a spike event or when the learning/inference task is completed. We have added the lines ‘184-185’ to increase clarity in the revised manuscript:

‘The proposed circuit is event driven and all transistors except MP2, MP3, MN2 are turned OFF when the threshold voltage saturates (i.e. once the baseline voltage is reached) in the absence of a spike event.’

(b) Some readers will be interested in why 2V Vdd is used in IDLE state and 3V otherwise.

We agree with the reviewer’s point. Actually, OxRAM device reliability is dependent on the voltage stress that the device undergoes. In the absence of spike events, the device remains in IDLE state for a larger duration. Here, we apply a reduced voltage of 2 V to reduce the stress on the device based on pulse optimization achieved. The voltage should also be small enough to ensure that the resistance state of the device is not disturbed while in IDLE state.

We have added following lines ‘185-187’ in revised manuscript to clarify this:

‘A reduced supply voltage is applied on the device in IDLE mode to reduce the stress on the device. Further, a smaller voltage also ensures that the resistance state of the device is not disturbed while in IDLE state.’

3. The authors' experimental setup is shown in Fig. S3. Information related to fabrication process of the CMOS circuit would be required.

Response R3.3: These are large commercial enhancement type HV CMOS devices with ~ 5um length. Large CMOS devices had to be used as selectors for the OxRAM devices as the OxRAM required high drive current due to their large dimension (50 um x 50 um), owing to limitations of the lithography node of the university fabrication facility. As suggested by the reviewer, details have now been added in the methods section of the revised manuscript.

4. In Fig. 6(b), the error bar for ΔI is significantly larger than that for the others. Discussion on this point would extend profoundness of the manuscript.

Response R3.4:

We thank the reviewer for pointing out this detail. We went through our experimental data and found that for most cycles ΔI falls in the same range as mentioned below. However, for a couple of cycles it shows larger changes which makes the overall error bar larger. This can be attributed to the fact that the jump ΔI basically depends on the SET/RESET switching and the resulting OxRAM device resistance. Since, the switching process is inherently stochastic some cycles resulted in larger changes making the entire error bar look larger.

Cycle	C1	C2	C3	C4	C5	C6	C7	C8	C9	C10
Sequence S1, ΔI (Volts)	0.56	0.58	0.58	0.58	0.72	0.62	0.74	0.68	0.62	0.56

5. In lines 212-213, it is written that Figs. 7(b) and (c) are from a random population during the learning process.

(a) In Figs. 7(b) and (c), 120 and 100 neurons are plotted respectively. So, it seems that these plots are from all the neurons.

Response R3.5:

We thank the reviewer for pointing this. In Fig. 7(b) and (c) the spike rasters are actually plotted for all the regular and adaptive neurons. For clarifying this we have corrected it and added lines 228-229 in revised manuscript as

‘Fig. 7(b) and Fig. 7(c) shows the spike rasters of LIF and DEXAT neurons respectively used in the network during the learning process.’

(b) It would be more clear if what the colors (black and red) of the dots mean in Fig. 7(b).

We thank the reviewer for this suggestion. We admit that the colour differentiation was done unintentionally on our part while plotting these graphs. We did not intend to show any differentiation based on colours in Fig. 7(b). All the neurons in Fig. 7 (b) denote LIF neurons. **We have now denoted all the spike rasters by black dots in the revised manuscript.**

(c) The title of Fig. 7 expresses that the plots are obtained after the training process. Which is correct?

The reviewer is right in pointing out that the plots in Fig.7 are obtained for inference (i.e. after training) of an MNIST handwritten image presented sequentially over a period of time.

Details are as mentioned in Methods section under Simulations heading in revised manuscript:

‘All the 60,000 images in the train set of MNIST database are used for training. For inference, input test image is presented from 10,000 test images in a sequential manner where each input pixel is presented in 1 time step ‘dt’.’

6. In lines 232-234, the authors argue that the effect of fabrication variability can be evaluated by inserting noise. Though the authors cite references, because it is not very intuitive, explaining the reason would strongly help readers' understanding.

Response R3.6:

We thank the reviewer for this suggestion. To avoid confusion we have removed the word ‘noise’ from the description. Further we have added a detailed sub-section on resultant variability calculation in the Methods section of the revised manuscript. Moreover, Supplementary Fig. 6 and Supplementary Fig. 7 have also been added to show detailed variability measurements and simulated neuron traces for the full networks.

7. As shown in Fig. 8, performance of this work does not exceed that of LSTM. A strong driving force for studying spiking system is energy efficiency as the authors write in line. 17. Thus I believe it is imperative to discuss energy consumption of the proposed system. Power consumption of the circuit could be added in Table 2 and compared with other works.

Response R3.7:

We thank the reviewer for raising this important point. For the revised manuscript we performed energy/power estimation for the threshold modulation block of the proposed DEXAT neuron and tried to link it with LSNN inference. Firstly, we would like to point out that energy dissipation in our proposed adaptive neuron circuit is not constant, as it depends on the exact number of the spike after the onset of neuron activity. In order to estimate the dissipation, we first experimentally measure the current flowing through the RRAM device in our circuit for any activity sequence (example - S1 is shown in Fig. R15). Current during SET is larger than RESET. Instantaneous current value shown in Fig. R15 is multiplied with supply voltage to get instantaneous power values. The instantaneous power values are integrated over the time duration (i.e. $\text{Energy (E)} = \int P_{\text{inst}} dt$) of sequence to get energy consumed in the complete spike sequence. One can empirically correlate the consecutive increments in threshold voltage (Δ) with the corresponding increase in current flowing in the circuit (increases with each consecutive SET pulse). Magnitude of increment of threshold voltage is directly proportional to the magnitude increment of average SET current for consecutive spikes. Hence, energy values for an arbitrary firing activity of DEXAT neuron can be estimated in the above way. Table R2 shows estimated “energy/per-inference” for threshold-modulation block of the proposed DEXAT neuron for (i) Speech recognition and (ii) SMNIST classification applications. For SMNIST classification using DEXAT based LSNN (100 DEXAT neurons) 60000 images were used for full dataset inference. Estimated inference energy dissipation/per DEXAT for each of the 60000 inferences are shown in Figure R16. It can be observed that energy values are scattered over a range. This can be attributed to the instantaneous firing patterns and activity of the hidden layer neurons.

Figure R15. Instantaneous current in the circuit during application of *SI*. Current flowing through the threshold modulator circuit during application of SET and RESET pulses in spike events for V_{DD_SET} of 3.5 V and V_{DD_RESET} of 3 V.

Figure R16. Energy consumption per inference task for SMNIST task. 60000 sample images are used for inference. Average DEXAT threshold modulation energy per neuron in a single inference is calculated and plotted for each inference task involving 100 DEXAT neurons in the hidden layer of LSNN.

To account for the activity dependent energy dissipation nature of DEXAT neurons, it is essential that energy be reported in context of specific applications. Thus, in Table R2 we report maximum, median and minimum energy per inference for SMNIST and speech recognition applications for the DEXAT LSNNs shown in the manuscript. In Table R2, we compare with literature reported dissipation values for other relevant adaptive neuron circuits. The Ni/HfO₂/Al doped TiO₂/TiN based circuit, characterized in this work, has an OxRAM device crosspoint dimension of 50 μm and CMOS selector of 5 μm length (@ 5V VDD), thus its energy consumption is in the order of ~ microjoules. Further, in Table R2 we estimate projected dissipation of DEXAT threshold modulator block when scaled OxRAM devices are used. A 500 nm OxRAM device [28] results in ~ 27 X reduction, while a 30 nm device [28] results in ~2292 X reduction, in energy respectively, compared to the 50 μm device.

Further, we perform a detailed performance (power, energy, area) benchmarking across multiple simulated CMOS technology nodes (180 nm, 90 nm, 65 nm, 28 nm) and FPGA for digital implementations of our proposed DEXAT neuron (Table R3). Description of DEXAT neuron threshold modulator block is first defined in Verilog HDL. Next, digital ASIC DEXAT threshold modulator block is synthesized and simulated using multiple CMOS technology node libraries in Cadence Encounter tool to estimate approximate power dissipation and area. It is important to note that the digital blocks are as-synthesized and not further optimized. We observed a clear power and area scaling trend with the node till 28 nm. The verilog implementation is also

synthesized on a Xilinx Artix 7 FPGA board using Xilinx Vivado tool to estimate power consumption for different precision. As expected, power and area increase with precision due to increased resource utilization on the FPGA. Energy consumption of digital threshold modulator block for SMNIST and speech recognition tasks was found to be in the order of ~ nanojoules and even ~ picojoules for some cases. From table R3, area overhead for digital implementations was found to be much larger compared to equivalent OxRAM based circuits.

Table R2: Estimated energy benchmarking with other relevant adaptive neurons.

Ref.	Neuron Type and Technology	Energy/Power	Remarks	Method	
[21]	-Mihalas Neibur model, Single exponential -0.5 um CMOS based Adaptive circuit	40 nW @ 5V (for threshold modulation block)	-no network application shown - fabricated	Simulated energy values per-spike (Reported in literature)	
[22]	ADeX Model, Single exponential 65 nm CMOS based Adaptive circuit	200 pJ @ 300Hz firing for (full neuron circuit)	- no network application shown - 32 neuron array fabricated		
[23]	-RRAM based Adaptive circuit, -Single exponential 65 nm CMOS + RRAM	Not reported	-simulated adaptive neuron circuit -MNIST using SNN	Not reported	
This Work	- DEXAT Model, Double exponential -Ni/HfO ₂ /Al doped TiO ₂ /TiN OxRAM -50 um cross point OxRAM + -0.5 um CMOS circuit -Energy reported for threshold modulation block	1.75 mJ (Max)	Energy per inference for SMNIST classification	Measured	
		298μJ (Median)			
		10μJ (Min)			
		696μJ (Max)			Energy per inference for Speech recognition
		116 μJ (Median)			
		13 μJ (Min)			
	-DEXAT Model, Double exponential -Mo/TiOx/TiN device [500 nm cross point RRAM] -Energy reported for threshold modulation block	58 μJ (Max)	Energy per inference for SMNIST classification	Projected using values from [28]	
		11.12μJ(Median)			
		1.85μJ (Min)			
		22.27 μJ (Max)	Energy per inference for Speech recognition		
		3.57 μJ (Median)			
		0.24 μJ (Min)			
	-DEXAT Model, Double exponential -Mo/TiOx/TiN device [30 nm cross point RRAM] -Energy reported for threshold modulation block	0.77 μJ(Max)	Energy per inference for SMNIST classification		
		0.13μJ (Median)			
		3.38 μJ (Min)			
0.3 μJ (Max)		Energy per inference for Speech recognition			
46.2 nJ (Median)					
0.4 nJ (Min)					

Table R3: Power, Energy, Area estimation of digital DEXAT implementation for different technology nodes.

Type of Digital Implementation & foundry	Precision	Power	Avg. Energy/per Inference - SMNIST Classification (J)	Avg. Energy/per Inference - Speech Recognition (J)	Area (μm^2)
ASIC @ SCL180 nm (simulated)	16 bit fixed point	1.24 mW	8.68E-08	1.14E-08	82658.5
	Floating point	12.45mW	8.72E-07	1.14E-07	675717.9
ASIC @ UMC 90nm (simulated)	16 bit fixed point	0.11 mW	7.70E-09	1.01E-09	32586
	Floating point	2.10 mW	1.47E-07	1.93E-08	397007
ASIC @ TSMC 65nm (simulated)	16 bit fixed point	45.19 μW	3.16E-09	4.14E-10	11435.7
	Floating point	1.24 mW	8.68E-08	1.14E-08	88377.1
FPGA (Xilinx Artix 7 xc7a35tcbg236) (28 nm)	16 bit fixed point	453 μW	3.17E-08	4.15E-09	-
	Floating point	4.63 mW	3.24E-07	4.24E-08	-

An approximate and qualitative benchmarking of proposed circuit with measured and projected values is shown in Fig. R17 comparing various types of realizations (digital, memristive, analog) for the adaptive threshold modulator block.

Figure R17. Approximate benchmarking of threshold modulator block with other adaptive threshold neuron functions: Power vs Area comparison for various realizations of DEXAT threshold modulator block (digital, OxRAM based) w.r.t closest analog CMOS adaptive neuron circuits [21, 22] from literature. Error bar indicates best case and worst case power values for OxRAM based circuits (blue squares) for the SMNIST application.

The comparison in Fig. R17 has inherent limitations as works from literature [21, 22] implement only single exponential function, and report only a single energy/power value (without reporting max, min). Still this

is the most justified benchmarking that we can come up with based on reviewer requests. Clearly from Fig. 17, in case of OxRAM based circuits, there is a clear power-area scaling trend visible w.r.t technology node. Thus one can deduce that for scaled nodes (i.e. 28 nm and below) OxRAM based DEXAT circuit (memristive circuits) have a clear advantage compared to other state-of-the-art implementations.

Performance benchmarking (power/energy/area) is discussed in Supplementary note 4 in revised manuscript, Supplementary Table 3 and Supplementary Fig. 12 are also added in revised manuscript.

Minor points

1. It would improve clarity of the manuscript if overall equations of the neuron and synapse models are presented in supplementary materials.

As per reviewer suggestion we have rectified this.

2. In line 66, reference No.24 is cited. But I could not understand how it differs from reference No.4. In addition, it seems that no explanation for the STORE and RECALL task is given in both references. The authors might mistakenly cited No.24.

Added a proper reference now.

3. Figure 4(a) would be more precise if a reset signal from Pulse Generator to Integrator is drawn.

Done.

4. In Fig. 8(a), it might be more clear if LSNN is explained as ALIF-based LSNN.

Done

Reviewers' Comments:

Reviewer #1:

Remarks to the Author:

This revised manuscript is significantly improved over the previous version. The advances beyond previous works have been better articulated, and the spoken voice command recognition task is more suitable for this LSNN network. I recommend publication of this work after addressing the following remaining concerns.

1. The author demonstrates the advantages of memristor in implementing the threshold adaptation module in terms of area and power consumption. Another important metric that needs to be discussed is the device endurance, because RRAM needs to be continuously programmed in the proposed 6T1R circuit which appears to have higher requirements for the endurance of RRAM compared to the case when it is used as artificial synapse. Whether the devices can maintain the original electrical behavior after the system works for a long time? The authors should further evaluate the hardware system's requirements for device endurance.

2. The authors have added the spoken voice command recognition task in the revision and the experimental results show that the system has a good performance. However, the authors did not give a specific recognition rate or accuracy which is an important indicator to describe the system performance and compare with other works.

Reviewer #2:

Remarks to the Author:

In the revised manuscript, the authors have demonstrated a full end-to-end LSNN with hardware CMOS-OxRAM DEXAT neurons integrated in real-time. They designed and fabricated a suite of experimental test benches. These results have shown the speech recognition process using the DEXAT neurons is efficient. With the new-added experiments, this paper demonstrates the novelty of the DEXAT neurons for spiking neural network applications.

Reviewer #3:

Remarks to the Author:

Please refer to the attached pdf file.

--

R3.1

Thank you for your kind explanation. Now I understand the definition of ρ_j is $\exp(-\delta t/\tau_a)$. From Eq. (2), following equation is obtained.

$$b_j(t+\delta t) - b_j(t) = (1 - \rho_j)(z_j(t) - b_j(t)).$$

Thus, if $\rho_j \stackrel{\text{def}}{=} \exp(-\delta t/\tau_a)$, the increase of the threshold voltage when the neuron fires becomes $(1 - \exp(-\delta t/\tau_a))(z_j(t) - b_j(t))$. But, the authors wrote that the increase is β/τ_a . For this to be correct, it seems that all of the following conditions have to be satisfied.

- a) $\delta t/\tau_a$ is sufficiently small (as the authors replied in R3.1(a))
- b) $b_j(t) = 0$ when spike arises (maybe b_{j_0})
- c) $z_j(t) = 1/\delta t$ (maybe $1/\delta t + b_{j_0}$) for the time step when spike arises and $z_j(t) = 0$ (maybe b_{j_0}) otherwise.

Is this understanding right? If right, please explain these points explicitly in the main text. Especially, c) is not common for readers outside the field of spiking neural networks ("the output spike of the neuron" is not sufficient). So, explicitly explaining them would be very helpful.

R3.2, R3.3, R3.5, R3.6

The points have been cleared.

R3.4

Thank you for your explanation. I understand that the variance may be reduced by improving the switching circuit.

R3.7

Overall direction of the energy dissipation evaluation is very good.

(1) In Supplementary note 4, the authors write that the estimated values in Supplementary Table 2 are that for the threshold-modulation block. Is the power consumed in other blocks (Pulse generator, Integrator, and Comparator) ignorable? I believe it is important to explain the reason why the power in these blocks can be ignored. If it is not ignorable, showing the value is imperative. Though these blocks are not the main part of this work, the value in other works in the table is for the whole neuron circuit.

(2) Power consumption of the authors' circuit seems to be higher than the other works. Evaluating power with scaled OxRAM is a good idea, but the values seem to be still high. It would be a good idea to explain some directions to reduce the power dissipation in the future (maybe in Conclusion section).

Point by Point Response Sheet

Reviewer #1

This revised manuscript is significantly improved over the previous version. The advances beyond previous works have been better articulated, and the spoken voice command recognition task is more suitable for this LSNN network. I recommend publication of this work after addressing the following remaining concerns.

We thank the reviewer for appreciating our revised manuscript. We have addressed the remaining concerns as detailed below:

1. The author demonstrates the advantages of memristor in implementing the threshold adaptation module in terms of area and power consumption. Another important metric that needs to be discussed is the device endurance, because RRAM needs to be continuously programmed in the proposed 6T1R circuit which appears to have higher requirements for the endurance of RRAM compared to the case when it is used as artificial synapse. Whether the devices can maintain the original electrical behavior after the system works for a long time? The authors should further evaluate the hardware system's requirements for device endurance.

Response R1.1

- a) We agree with the reviewer's concern regarding the RRAM device endurance. We have acknowledged this limitation in lines '251-254' of the revised manuscript as quoted below:

"It is important to note that OxRAM device will undergo programming (SET/RESET) during the inference process. This imposes high endurance requirements on the OxRAM devices being used to realize the neuron circuit. In supplementary note 5, we present a detailed analysis on the impact of OxRAM device endurance (resistance window degradation) on overall network inference accuracy"

- b) Further based on the reviewer's suggestion we performed a detailed analysis of '**network performance vs RRAM device endurance**' with the help of additional simulations and empirical modeling of device resistance window degradation with cycling. A new section Supplementary Note 5 on endurance analysis is added in the revised supplementary material sheet.

Key observations from endurance analysis are outlined below:

1. As per our analysis, if RRAM devices have endurance $< 10^6$ cycles then it is preferable to train the network offline using the proposed DEXAT model and use RRAM based low-power deployment on full hardware for inference tasks. Depending upon the device endurance and application at hand, some minor in-situ training may still be possible on hardware.
2. Training can also be done using digital implementations such as state of the art FPGAs as they will not pose an endurance issue.
3. RRAM for in-situ training is possible with the use of state-of-the-art bi-layer RRAM devices. Some devices have been reported in literature [a], [b] having endurance as high as 10^{12} with stacks similar to the one used in this study.
4. We estimated the statistics (min, max, mean) for programming cycles (SET/RESET) that an OxRAM device inside the hidden layer DEXAT neuron would undergo during inference for all 3 datasets presented

in the paper (SMNIST, speech recognition, and store-recall). These cycling statistics are shown below in Table R1.

5. Table R1 also explores the impact of network size on device cycling statistics. **It was observed that as network size increases mean cycling events per device decrease. Hence, the endurance requirement per device in the hidden layer tends to relax.** Thus, larger hidden layers can help to bring down individual device stress for specific applications.
6. Further, based on device cycling statistics we estimate the number of inferences (for each application) that a full hardware network can perform before reaching the endurance limit. This analysis was done assuming networks built from RRAM devices of 3 different flavors, spanning an endurance range of 6 orders of magnitude (Device 1: 10^6 cycles, Device 2: 10^9 cycles, and Device 3: 10^{12} cycles).
7. Further, we also model the impact of device resistance window degradation with cycling on network performance. As RRAM devices cycle more and more, the resistance window tends to squeeze. Authors in [c] studied the effect of endurance degradation on analog RRAM devices. Using this study [c] as the basis we modeled R_{on} / R_{off} ratio degradation to carry out dynamic LSNN inference simulations where DEXAT neuron behavior changes actively with the number of cycles. Modeled degradation of DEXAT neuron threshold voltage curves with cycling is shown in Fig. R1 and Fig. R2. Next, we inject these new degraded DEXAT curves dynamically in the network based on inference cycle/neuron firing activity count and re-estimate the network accuracy. Table R2 shows the effect of active degradation of resistance window on inference accuracy. **We observe that the network is able to maintain a sufficiently high accuracy even for $\sim 44\%$ drop in the original R_{on} / R_{off} ratio.**

Figure R1: Dynamic modeling of degradation of adaptive threshold decay behaviour of our DEXAT neuron as the device R_{on}/R_{off} ratio degrades with increasing number of programming cycles. Scaling of R_{on}/R_{off} ratio is based on characterization shown in [d]. Note overall window squeezes with cycling.

Figure R2: Simulated DEXAT neuron behaviour for multiple spike events after injecting dynamic cycling/lifetime based threshold decay curves for respective DEXAT hidden layer neurons in LSNN simulations. After a large number of cycles almost no DEXAT action is observed.

Table R1: Estimation of programming cycle requirement/ per DEXAT neuron depending on network size and application. Table also shows the approximate number of inferences before device breakdown.

Application (Dataset)	Network Size (LIF-DEXAT)	Cycling event statistics/per inference (for hidden layer DEXAT neurons)			No. of inferences that can be supported* (with different ideal device endurance)		
		Max	Min	Mean	10 ⁶ cycles	10 ⁹ cycles	10 ¹² cycles
SMNIST (Statistics on 10,000 test dataset inferences)	12-10	99	0	60	~ 10.1 x 10 ³	~10.1 x 10 ⁶	~10.1 x 10 ⁹
	30-25	87	0	31	~ 11.5 x 10 ³	~11.5 x 10 ⁶	~11.5 x 10 ⁹
	120-100	61	0	20	~ 16.4 x 10 ³	~16.4 x 10 ⁶	~16.4 x 10 ⁹
Speech (2 class) (Statistics on 734 test dataset inferences)	10-10	58	0	34	~ 17.2 x 10 ³	~17.2 x 10 ⁶	~17.2 x 10 ⁹
	50-50	11	0	2	~ 90.9 x 10 ³	~90.9 x 10 ⁶	~90.9 x 10 ⁹
Speech (12 class) (Statistics on 4890 test dataset inferences)	100-100	486	0	86	~2.1 x 10 ³	~2.1 x 10 ⁶	~2.1 x 10 ⁹
	300-300	434	0	79	~2.3 x 10 ³	~2.3 x 10 ⁶	~2.3 x 10 ⁹
	500-500	445	0	52	~2.2 x 10 ³	~2.2 x 10 ⁶	~2.2 x 10 ⁹
STORE-RECALL	10-10	39	0	16	~ 25.6 x 10 ³	~25.6 x 10 ⁶	~ 25.6 x 10 ⁹
* Number of supported inferences are calculated assuming that each device undergoes the 'max' (i.e. worst case) number of cycling events in each inference.							

Table R2: Test accuracy degradation with cycling for SMNIST task (each run corresponds to inference on 10000 test images using LSNN with 120 LIF and 100 DEXAT neurons).

Cycles (number of programming hits taken by the device during inference)	< 10 ⁶ cycles	10 ⁶ -10 ⁷ cycles	10 ⁷ -10 ⁸ cycles	10 ⁸ -10 ⁹ cycles	> 10 ⁹ cycles
(a) *Drop in resistance window (Ron/ Roff) %	0 %	11 %	44 %	54 %	74 %
(i) Test Accuracy (%) [with resultant variability = 10%]	95.4 %	94.6 %	87.5 %	77.9 %	43 %
(ii) Test Accuracy (%) [with resultant variability = 30%]	93.2 %	92.6%	80.9 %	70. %	37.6%
(b) **Drop in resistance window (Ron/ Roff) %	0 %	20 %	40 %	60%	80 %
Test Accuracy (%) [Resultant variability = 30%]	93.2 %	92.3%	88.5 %	62.5%	39.7%

*In this case RRAM resistance window degradation with cycling corresponds to data extracted from ref [c].

**In this case RRAM resistance window degradation with cycling is assumed to be uniform 20% in each window.

References for R1.1:

[a] Hsu et al. "Self-rectifying bipolar TaO_x/TiO₂ RRAM with superior endurance over 10¹² cycles for 3D high-density storage-class memory." *2013 Symposium on VLSI Technology*. IEEE, 2013.

[b] Lee et al. "A fast, high-endurance and scalable non-volatile memory device made from asymmetric Ta₂O_{5-x}/TaO_{2-x} bilayer structures." *Nature materials* 10.8 (2011): 625-630.

[c] Zhao et al. "Characterizing endurance degradation of incremental switching in analog RRAM for neuromorphic systems." *2018 IEEE International Electron Devices Meeting (IEDM)*. IEEE, 2018.

2. The authors have added the spoken voice command recognition task in the revision and the experimental results show that the system has a good performance. However, the authors did not give a specific recognition rate or accuracy which is an important indicator to describe the system performance and compare with other works.

Response R1.2

We understand the reviewer's concern. We have now included accuracy values and benchmarking w.r.t state-of-the-art for the speech recognition task in the revised manuscript (lines '256-261') and Supplementary Figure 15. We performed multiple simulations on the speech recognition task (GSC-dataset) using different network sizes, based on hardware extracted (HfO₂ / TiO₂ OxRAM device) parameters as shown in Fig. R3. We performed simulations for both ALIF and DEXAT based LSNN to compare. As reported in literature, state-of-the-art speech recognition results on GSC dataset with ALIF based LSNN required a network size of 2048 hidden layer neurons [d] to achieve an accuracy of ~91 %. **We were able to achieve similar accuracy using our proposed DEXAT based LSNN with ~ 51% less hidden layer neurons (see Fig.R3 (a)).** Using 300 LIF + 300 DEXAT neurons our

simulated LSNN could achieve an accuracy of $\sim 90\%$, i.e. $\sim 2.5\%$ higher than the same sized ALIF based LSNN, in much fewer iterations (see Fig.R3(b)).

Figure R3 LSNN classification performance on GSC dataset. (a) Test accuracy vs network dimension. DEXAT based LSNN achieves higher accuracy compared to ALIF based LSNN even for significantly fewer hidden layer neurons. (State-of-the-art-accuracy [d] is achieved even with $\sim 51\%$ lesser neurons). All binary class simulations are performed after training the LSNN on a reduced 2-class GSC dataset. DEXAT binary class result for network size (10-10) corresponds to the end-to-end speech experiment described in manuscript. Also, it can be seen that accuracy increases with increasing network size for both binary and 12-class GSC. **(b)** Test accuracy comparison for a 300-300 sized ALIF vs DEXAT LSNN. For all GSC simulations, DEXAT neurons are used based on the hardware extracted parameters on $\text{HfO}_2/\text{TiO}_2$ device.

Reference for R1.2:

[d] Salaj, Darjan, et al, "Spike-frequency adaptation provides a long short-term memory to networks of spiking neurons." *bioRxiv* (2020)." *bioRxiv* (2020).URL:

<https://www.biorxiv.org/content/10.1101/2020.05.11.081513v1.abstract>

Reviewer #2 (Remarks to the Author):

In the revised manuscript, the authors have demonstrated a full end-to-end LSNN with hardware CMOS-OxRAM DEXAT neurons integrated in real-time. They designed and fabricated a suite of experimental test benches. These results have shown the speech recognition process using the DEXAT neurons is efficient. With the new-added experiments, this paper demonstrates the novelty of the DEXAT neurons for spiking neural network applications.

We thank the reviewer for appreciating our new results on full experimental end to end demonstration of LSNN. We also thank the reviewer for earlier useful comments which helped us to enhance the manuscript.

Reviewer #3

R3.1 Thank you for your kind explanation. Now I understand the definition of ρ_j is $\exp(-\delta t/\tau_a)$. From Eq. (2), following equation is obtained. $b_j(t+\delta t) - b_j(t) = (1 - \rho_j)(z_j(t) - b_j(t))$. Thus, if $\rho_j \approx \exp(-\delta t/\tau_a)$, the increase of the threshold voltage when the neuron fires becomes $(1 - \exp(-\delta t/\tau_a))(z_j(t) - b_j(t))$. But, the authors wrote that the increase is β/τ_a . For this to be correct, it seems that all of the following conditions have to be satisfied.

- a) $\delta t/\tau_a$ is sufficiently small (as the authors replied in R3.1(a))
- b) $b_j(t) = 0$ when spike arises (maybe $b_j = 0$)
- c) $z_j(t) = 1/\delta t$ (maybe $1/\delta t + b_j = 0$) for the time step when spike arises and $z_j(t) = 0$ (maybe $b_j = 0$) otherwise.

Is this understanding right? If right, please explain these points explicitly in the main text. Especially, c) is not common for readers outside the field of spiking neural networks (“the output spike of the neuron” is not sufficient). So, explicitly explaining them would be very helpful.

Response R3.1:

We understand the reviewer’s point and try to explain it below. As pointed out by reviewer increase in threshold voltage can be written as:

$$b_j(t+\delta t) - b_j(t) = (1 - \rho_j)(z_j(t) - b_j(t)) \text{ -----(a)}$$

Putting $\rho_j = \exp(-\delta t/\tau_a)$ in (a)

$$b_j(t+\delta t) - b_j(t) = (1 - \exp(-\delta t/\tau_a))(z_j(t) - b_j(t)) \text{ -----(b)}$$

Now as reviewer correctly pointed out if conditions (a) and (c) (i.e. τ_a is large implying $\delta t/\tau_a$ is small and $z_j(t) = 1/\delta t$) are satisfied equation (b) reduces to

$$b_j(t+\delta t) - b_j(t) = (\delta t/\tau_a)((1/\delta t) - b_j(t))$$

$$b_j(t+\delta t) - b_j(t) = 1/\tau_a - (\delta t/\tau_a)b_j(t) \text{(c)}$$

Multiplying by β on both sides of equation (c)

$$\beta \{b_j(t+\delta t) - b_j(t)\} = (\beta/\tau_a) - \beta(\delta t/\tau_a)b_j(t) \text{(d)}$$

Now, condition (b) as pointed out by reviewer (i.e. $b_j(t) = 0$) would be true for the very first spike event for which neuron threshold voltage is at baseline b_{j0} . Hence, in that case threshold jump (Δ) at spike event clearly equals β/τ_a . For any successive spike event, $b_j(t)$ would be a small non-zero value representing the decayed value of adaptive threshold at time ‘t’. However, since τ_a is large (i.e. $\delta t/\tau_a$) is very small so the term $\{\beta(\delta t/\tau_a)b_j(t)\}$ in equation (d) tends to zero and effective threshold jump (Δ) at spike event is still β/τ_a .

On reviewer’s request we have now added the necessary details in our revised manuscript in lines ‘57-58’.

R3.7 Overall direction of the energy dissipation evaluation is very good. (1) In Supplementary note 4, the authors write that the estimated values in Supplementary Table 2 are that for the threshold-modulation block. Is the

power consumed in other blocks (Pulse generator, Integrator, and Comparator) ignorable ? I believe it is important to explain the reason why the power in these blocks can be ignored. If it is not ignorable, showing the value is imperative. Though these blocks are not the main part of this work, the value in other works in the table is for the whole neuron circuit. (2) Power consumption of the authors' circuit seems to be higher than the other works. Evaluating power with scaled OxRAM is a good idea, but the values seem to be still high. It would be a good idea to explain some directions to reduce the power dissipation in the future (maybe in Conclusion section).

Response R3.7:

(1) We thank the reviewer for these suggestions. The reviewer is right in pointing out that we have reported the energy/power consumption only of our threshold modulator block in Supplementary Table 2. However, we would like to mention that among the two references that we used for comparison, for reference [21] we have reported the power of **only threshold modulator and membrane circuits (excluding the comparator and pulse generator)**. We acknowledge that for the other reference [22] the energy/power reported in Table 2 is of the complete neuron circuit. As authors in [22] report only combined power, thus we are unable to isolate it. For sake of clarity and fairness we have acknowledged this difference in the footnote below the Supplementary Table 2 in the revised Supplementary sheet. Further, as per reviewer suggestion, we surveyed existing literature for isolating power/energy consumption of other neuron blocks (i.e. non threshold modulator blocks). It is difficult to separately extract the power/energy of the individual blocks as in most cases, the dissipation is reported for full circuits.

However, we observed that the total neuron power consumption for many full neuron circuit implementations like [e]-[g] is of the order of a few 10's of uW (@ CMOS nodes of 500nm, 350nm). Correspondingly, we note that estimated power consumption for our DEXAT threshold modulator block (projections on comparable 500 nm devices), ranges from ~2uW to ~24uW as shown in Fig.R4. Further, we also observe that for neuron in [22] (@ 65nm CMOS node) total neuron circuit average power consumption is 60 nW. Our projection of power dissipation for the DEXAT threshold modulator block on the closest comparable scaled device (i.e. 28nm RRAM) lies in the range of ~3.8nW to ~320nW.

Thus, from these observations, we feel that the total power consumption of a full neuron circuit involving (i) our threshold modulator block and (ii) any other state-of-the-art non-threshold neuron block circuit, reported in literature would be of similar order (as full reported circuits) and will not drastically shoot up or change by an order of magnitude. In other words, what we are trying to say is that even though we report only threshold modulator block circuit dissipation, however the dissipation of other blocks is more or less in the same order, so the two dissipations at worst would linearly add up 2X and not significantly jump by an order of magnitude or more. Further, as already acknowledged in the manuscript we have implemented our non-threshold blocks in software and external PCB board. Thus, it will not make sense to report the dissipation of blocks implemented in software and discrete PCB components. True or representative circuit dissipation values would be the ones arising from a single ASIC/design. Due to practical limitations and scope of the task we cannot realize a full ASIC in this work. Thus, we kindly request the reviewer to consider our reasoning for focusing and reporting mainly the threshold modulator block dissipation.

(2) Reviewer is correct that we have projected power based on a scaled OxRAM device. We would like to add that as shown in the Fig. R4 , the median power of our threshold block is higher than [21]-[22], however since we are reporting the statistics on 60,000 inference runs, we observed a long dispersion bar (min, max) on this value and for several instances our power dissipation was lower than the one in [21]-[22] as shown in Fig.R4. Since such statistical data is not available for [21]-[22] we did not explicitly mention this gain in earlier versions of the manuscript. Based on reviewer suggestion, we think following two future directions can further increase the efficiency: (i) minimizing individual switching-event power/energy through materials engineering, ie. low

programming voltages, smaller switching currents (preferably in \sim nA), ultra-fast switching speed (preferably in \sim ns) leading to sub-picojoule energies ($<$ 1pJ). (ii) Use of system level optimizations that may help to bring down individual neuron spiking activity without compromising the accuracy. As suggested by the reviewer we have added a few lines (quoted below) on possible future direction in the conclusion section of the paper (lines ‘290-292’).

“Future efforts should be in the direction of reducing individual OxRAM device switching energy to sub pico-joules i.e. through lower programming voltages, switching currents (\sim nA or lower), and ultra-fast switching speeds (sub ns).”

Figure R4 Approximate benchmarking of threshold modulator block with other adaptive threshold neuron functions. Power vs Area comparison for various realizations of DEXAT threshold modulator block (digital, OxRAM based) w.r.t closest analog CMOS adaptive neuron circuits [21, 22] from literature. Error bar indicates best case and worst case power values for OxRAM based circuits (blue squares) for the SMNIST application for 60,000 inference cases. Note that for multiple instances the power of OxRAM based system is lower than [22] and [21].

References for R3.7:

[e] van Schaik, André, et al. "A log-domain implementation of the Izhikevich neuron model." *Proceedings of 2010 IEEE International Symposium on Circuits and Systems*. IEEE, 2010.

[f] Wijekoon, Jayawan HB, and Piotr Dudek. "Compact silicon neuron circuit with spiking and bursting behaviour." *Neural Networks* 21.2-3 (2008): 524-534.

[g] Indiveri, Giacomo, Elisabetta Chicca, and Rodney Douglas. "A VLSI array of low-power spiking neurons and bistable synapses with spike-timing dependent plasticity." *IEEE transactions on neural networks* 17.1 (2006): 211-221.

Reviewers' Comments:

Reviewer #1:

Remarks to the Author:

The authors have addressed my final comments, and I would recommend publication of this work on Nature Communications.

Reviewer #3:

Remarks to the Author:

[R3.1]

Now it is clear. Thank you.

[R3.7]

(1) The authors' argument that the threshold modulation circuit's power consumption will be in the same scale as those in the state-of-the-art neuron circuits is reasonable. I believe it is better to include this argument in Supplementary note 4.

By the way, please note that [g] ([14] in the manuscript) is too old. A paper that covers their recent chip is:

Qiao et al., "A reconfigurable on-line learning spiking neuromorphic processor comprising 256 neurons and 128K synapses," *frontiers Neurosci*, 9, 141, 2015.

This paper is not really new, but it may be better to replace [14] with this.

(2) Thank you.

--

Point by Point Response Sheet for Manuscript NCOMMS-20-40665B

Reviewer #1 (Remarks to the Author):

The authors have addressed my final comments, and I would recommend publication of this work on Nature Communications.

Response: We thank the reviewer for acknowledging our revision and appreciating our work.

Reviewer #3 (Remarks to the Author):

[R3.7]

(1) The authors' argument that the threshold modulation circuit's power consumption will be in the same scale as those in the state-of-the-art neuron circuits is reasonable. I believe it is better to include this argument in Supplementary note 4.

By the way, please note that [g] ([14] in the manuscript) is too old. A paper that covers their recent chip is:

Qiao et al., "A reconfigurable on-line learning spiking neuromorphic processor comprising 256 neurons and 128K synapses," frontiers Neurosci, 9, 141, 2015. This paper is not really new, but it may be better to replace [14] with this.

Response:

We thank the reviewer for acknowledging our revision.

- a) Based on the reviewer's suggestion we have now added this argument in Supplementary note 4.

'Although, we report the power/energy only of the threshold modulator block (as other blocks are implemented in software), total energy/power of a neuron circuit built using proposed DEXAT adaptation block would be of the same order as that of individual DEXAT block and hence the comparison is valid.'

- b) We have also replaced ref. [14] in the manuscript with the more relevant reference suggested by the reviewer.